# Myosin II regulatory light chain phosphorylation and formin availability modulate cytokinesis upon changes in carbohydrate metabolism

Francisco Prieto-Ruiz[1], Elisa Gómez-Gil[1,2], Rebeca Martín-García[3], Armando Jesús Pérez-Díaz[1], Jero Vicente-Soler[1], Alejandro Franco[1], Teresa Soto[1], Pilar Pérez[3], Marisa Madrid[1]*, José Cansado[1]*

[1]Yeast Physiology Group. Department of Genetics and Microbiology. Campus de Excelencia Internacional de Ámbito Regional (CEIR) Campus Mare Nostrum, Universidad de Murcia, Murcia, Spain; [2]The Francis Crick Institute, London, United Kingdom; [3]Instituto de Biología Funcional y Genómica (IBFG), Consejo Superior de Investigaciones Científicas, Universidad de Salamanca, Salamanca, Spain

*For correspondence:
marisa@um.es (MM);
jcansado@um.es (JC)

Competing interest: The authors declare that no competing interests exist.

**Abstract** Cytokinesis, the separation of daughter cells at the end of mitosis, relies in animal cells on a contractile actomyosin ring (CAR) composed of actin and class II myosins, whose activity is strongly influenced by regulatory light chain (RLC) phosphorylation. However, in simple eukaryotes such as the fission yeast *Schizosaccharomyces pombe*, RLC phosphorylation appears dispensable for regulating CAR dynamics. We found that redundant phosphorylation at Ser35 of the *S. pombe* RLC homolog Rlc1 by the p21-activated kinases Pak1 and Pak2, modulates myosin II Myo2 activity and becomes essential for cytokinesis and cell growth during respiration. Previously, we showed that the stress-activated protein kinase pathway (SAPK) MAPK Sty1 controls fission yeast CAR integrity by downregulating formin For3 levels (Gómez-Gil et al., 2020). Here, we report that the reduced availability of formin For3-nucleated actin filaments for the CAR is the main reason for the required control of myosin II contractile activity by RLC phosphorylation during respiration-induced oxidative stress. Thus, the restoration of For3 levels by antioxidants overrides the control of myosin II function regulated by RLC phosphorylation, allowing cytokinesis and cell proliferation during respiration. Therefore, fine-tuned interplay between myosin II function through Rlc1 phosphorylation and environmentally controlled actin filament availability is critical for a successful cytokinesis in response to a switch to a respiratory carbohydrate metabolism.

## Editor's evaluation

This research advance manuscript breaks new ground by linking cytokinesis regulation to myosin light chain phosphorylation that is dependent on whether cells are growing through respiration or fermentation. This is an exciting new direction and will spur more activity in uncovering adaptations/ modifications of cytokinesis mechanisms to metabolic states.

## Introduction

Cytokinesis enables the physical separation of daughter cells after mitosis has been completed (*Green et al., 2012*). In non-muscle animal cells, this process relies on the formation of a contractile actomyosin ring ('CAR'), composed of actin filaments and myosin II (NMII), which provides the mechanical

force for actomyosin contractility (*Garrido-Casado et al., 2021*; *Mangione and Gould, 2019*). The prototype NMII is a complex assembled from two heavy chains, two essential light chains (ELC), and two regulatory light chains (RLC), which, in response to phosphorylation, cause NMII to fold into an extended and active conformation (*Garrido-Casado et al., 2021*). Phosphorylation of RLC at Ser19 is critical for NMII activation and results in the formation of bipolar filaments with increased actin binding affinity and ATPase motor activity (*Garrido-Casado et al., 2021*; *Craig et al., 1983*; *Trybus and Lowey, 1984*). Similar to its deletion or pharmacological inhibition (*Bao et al., 2005*; *Ma et al., 2010*), NMII is enzymatically inactive in the absence of RLC phosphorylation at Ser19 (*Trybus, 1989*), resulting in defective cytokinesis and an increased multinucleation (*Komatsu et al., 2000*). Several kinases are involved in RLC phosphorylation at Ser19 and NMII activation during cleavage furrow accumulation and CAR contraction during cytokinesis (*Garrido-Casado et al., 2021*). RLC phosphorylation at sites other than Ser19 provides additional layers of regulation for the positive or negative modulation of NMII contractile activity within specific cellular contexts (*Garrido-Casado et al., 2021*).

*S. pombe* is a Crabtree-positive fission yeast that grows through either fermentative or respiratory metabolism and is a well-established model organism for the study of cytokinesis (*Pollard and Wu, 2010*; *Rincon and Paoletti, 2016*; *Balasubramanian et al., 2004*). This simple eukaryote uses a CAR with two myosin-II heavy chains, Myo2 and Myp2/Myo3, to divide (*Wang et al., 2020*). Myo2 is essential for viability and cytokinesis during unperturbed growth, whereas Myp2 plays a non-essential but important role during CAR constriction, and in response to salt stress (*Laplante et al., 2015*; *Palani et al., 2017*; *Okada et al., 2019*). In contrast to NMII, Myo2 does not form filaments at physiological saline concentrations, but instead adopts a unipolar organization with head domains exposed to the cytoplasm and tails anchored in medial precursor nodes of the CAR at mitotic onset (*Pollard et al., 2017*; *Laporte et al., 2011*; *McDonald et al., 2017*; *Laplante et al., 2016*). The essential formin Cdc12 nucleates and elongates actin filaments at the nodes, whereas Myo2 promotes the fusion of the equatorial nodes to form a mature CAR (*Chang et al., 1997*; *Kovar et al., 2003*; *Vavylonis et al., 2008*). For3, a non-essential diaphanous-like formin that assembles actin cables for cellular transport, also plays an important role in nucleating actin filaments for the CAR during cytokinesis (*Coffman et al., 2013*; *Gómez-Gil et al., 2020*). In response to environmental cues and cytoskeletal damage, Sty1, a p38 MAPK ortholog and core effector of the SAPK pathway, blocks cell division by reducing For3 levels and the availability of actin filaments for the CAR (*Gómez-Gil et al., 2020*).

Cdc4 and Rlc1 are the respective fission yeast ELC and RLC shared by Myo2 and Myp2 (*Le Goff et al., 2000*; *Naqvi et al., 2000*; *McCollum et al., 1995*). Early evidence indicated that the p21/Cdc42-activated kinase (PAK) ortholog Pak1/Shk1/Orb2 phosphorylates Ser35 and Ser36 of Rlc1, which are homologous to RLC Thr18 and Ser19 in NMIIs (*Loo and Balasubramanian, 2008*). Phosphorylation of Rlc1 at Ser35 and Ser36 by Pak1 delayed cytokinesis, whereas expression of a non-phosphorylatable mutant (*rlc1-S35A S36A*), resulted in premature CAR constriction (*Loo and Balasubramanian, 2008*). This is consistent with in vitro data showing that Rlc1 phosphorylation reduces the interaction of Myo2 with actin during force generation (*Pollard et al., 2017*). This and subsequent work identified Ser35 as the sole target of Pak1 both in vitro and in vivo (*Pollard et al., 2017*; *Magliozzi et al., 2020*). However, another study described that the average in vitro motility rate of purified Myo2 bound to the Rlc1-*S35A S36A* mutant is reduced by ~25% compared to that of the myosin bound to wild-type Rlc1, and that phosphorylation at both sites is positive for CAR constriction dynamics (*Sladewski et al., 2009*). While the essential role of RLC phosphorylation for NMII activity is well established in animal cells, the biological significance of Rlc1 phosphorylation at Ser35 during cytokinesis remains unclear.

Here, we show that modulation of Myo2 activity by Rlc1 phosphorylation at Ser35 is essential for fission yeast cytokinesis and proliferation during respiratory growth. This modification is exerted by Pak1 together with Pak2, a second PAK ortholog whose expression increases during respiration. Rlc1 phosphorylation at Ser35 becomes essential due to the reduced availability of For3-nucleated actin filaments caused by SAPK activation during respiration-induced oxidative stress. Thus, formin-dependent actin filament nucleation and myosin II activity are coupled for optimal control of cytokinesis in response to changes in MAPK signaling and carbon source metabolism.

# Results

## Myosin II regulatory light chain phosphorylation is essential for *S. pombe* cytokinesis and growth during respiration

To gain further insight into the contribution of RLC phosphorylation to the myosin II-dependent control of cytokinesis in *S. pombe*, we expressed a C-terminal GFP-tagged version of Rlc1 under the control of its native promoter in *rlc1Δ* cells. This construct was fully functional and suppressed the defective CAR positioning and multiseptation associated with the lack of Rlc1 function (*Figure 1—figure supplement 1A*; *Le Goff et al., 2000*). The Rlc1-GFP fusion migrates as two distinct bands by Phos-tag SDS-PAGE analysis in extracts from exponentially growing cells (*Figure 1A*). The Rlc1 mobility of a mutant in which Ser36 was changed to alanine (Rlc1(S36A)-GFP) was similar to that of the wild-type. In contrast, only the faster-migrating band was present in mutants expressing Rlc1(S35A)-GFP or Rlc1(S35A S36A)-GFP fusions (*Figure 1A*), confirming that the slower-migrating band corresponds to Rlc1 phosphorylated at Ser35.

To precisely follow the dynamics of Rlc1 phosphorylation and localization during the cell cycle, we expressed the Rlc1-GFP version in cells carrying an analog-sensitive version of the Cdk1 kinase ortholog Cdc2 (*cdc2-asM17*) (*Aoi et al., 2014*) and a Pcp1-GFP fusion (pericentrin SPB component; internal control for mitotic progression). Simultaneous live fluorescence microscopy and Phos-tag SDS-PAGE analysis of synchronized cells released from the G2 arrest showed that in vivo Rlc1 phosphorylation at Ser35 was very low at nodes during CAR assembly, gradually increased during ring maturation, peaked at the onset of CAR contraction until the final stages, and slowly decreased slowly during septum closure and cell separation (*Figure 1B*). As previously suggested (*Loo and Balasubramanian, 2008*), these results confirm that in vivo Rlc1 phosphorylation at Ser35 is enhanced during CAR constriction and septum formation.

Time-lapse fluorescence microscopy of asynchronous glucose-growing cells revealed a minimal but statistically significant increase in the total time for ring constriction and disassembly in Rlc1(S35A)-GFP cells as compared to wild-type cells (18.36±2.06 *vs* 17.26±2.17 min, respectively), with a slightly slower ring constriction rate (0.595±0.02 *vs* 0.616+0.02 µm/min, respectively) (*Figure 1C*). Cells expressing a dual phospho-mimicking form of Rlc1 (Rlc1-S35D S36D) exhibit normal CAR dynamics and support cytokinesis like wild-type cells (*Sladewski et al., 2009*). Similarly, CAR dynamics and ring constriction rates were identical between wild-type and Rlc1(S35D)-GFP cells (*Figure 1C*). In contrast to non-muscle animal cells, where phosphorylation of the regulatory light chain is essential for NMII activity (*Garrido-Casado et al., 2021*), in vivo Rlc1 phosphorylation during *S. pombe* growth in the presence of glucose only minimally affects myosin II function for CAR dynamics.

These findings prompted us to search for other environmental and/or nutritional condition/s where Rlc1 phosphorylation-dependent control of myosin II activity might become essential for fission yeast cytokinesis. A recent study described that *rlc1Δ* cells struggle to grow in a glycerol-based medium, which imposes a respiratory metabolism (*Malecki et al., 2016*), as *rlc1Δ* cell growth was strongly reduced in plates containing 3% glycerol plus 0.08% glucose (*Figure 1D*). Strikingly, the unphosphorylated mutants *rlc1-S35A* and *rlc1-S35A S36A*, but not *rlc1-S36A* or the phosphomimetic versions *rlc1-S35D* and *rlc1-S35D S36D*, also grew very slowly in this medium (*Figure 1D*). This phenotype was dependent on Rlc1 phosphorylation at Ser35, as a conditional expression of an Rlc1-HA fusion in *rlc1Δ* cells by the β-estradiol-regulated promoter (*Ohira et al., 2017*) allowed their growth on glycerol, whereas conditional expression of the unphosphorylated Rlc1(S35A)-HA mutant did not (*Figure 1—figure supplement 1B*). *rlc1Δ* and *rlc1-S35A* cells on a prototrophic 972 hr background, but not those expressing the *rlc1-S35D* allele, were also growth defective on glycerol (*Figure 1—figure supplement 1C*), confirming that nutritional auxotrophic markers in the strains are not responsible for this phenotype.

In contrast to wild-type Rlc1, the growth of *rlc1-S35A* cells transferred to a glycerol-based liquid medium was limited to 3–4 additional divisions (*Figure 1—figure supplement 1D*), with a progressive increase in multiseptated cells with thickened septa and lysed cells, indicating the presence of a cytokinetic defect (*Figure 1E*, *Figure 1—figure supplement 1D*). Accordingly, the total ring assembly and contraction time in glycerol-incubated *rlc1-S35A* cells were much longer than in wild-type cells (61.67±7.36 *vs* 49.27±3.69 min, respectively) (*Figure 1F–G*). The cytokinetic delay was most pronounced during ring constriction and disassembly (34.00±6.16 *vs* 21.40±3.62 min, respectively), with a marked reduction in the ring constriction rate (0.384±0.02 *vs* 0.500+0.02 µm/min, respectively)

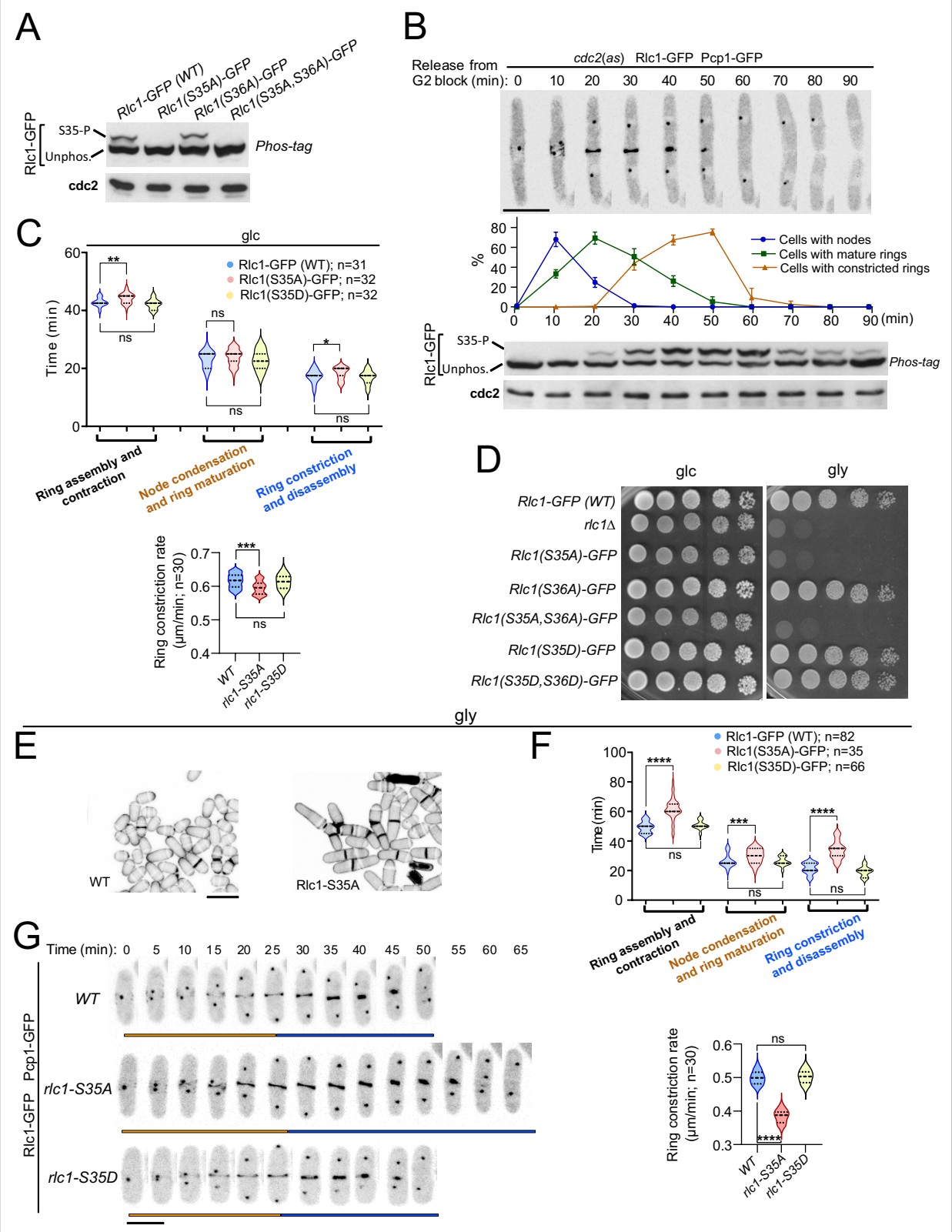

**Figure 1.** Myosin-II regulatory light chain phosphorylation is essential for *S. pombe* cytokinesis and growth during respiration. (**A**) Total protein extracts from the indicated strains grown exponentially in YES-glucose medium were resolved by Phos-tag SDS-PAGE, and the Rlc1-GFP fusion was detected by incubation with anti-GFP antibody. Anti-Cdc2 was used as a loading control. Rlc1 isoforms, phosphorylated (S35–P), and non-phosphorylated at Ser35 (Unphos), are indicated. The blot corresponds to a representative experiment that was repeated at least three times and the trend of the

*Figure 1 continued on next page*

*Figure 1 continued*

mobility shift was reproducible. (**B**) Cells with a *cdc2-asM17* analog-sensitive mutant allele expressing a Rlc1-GFP genomic fusion were arrested at G2 in YES-glucose medium supplemented with 3-NM-PP1 and incubated in the same medium without the kinase analog for the indicated times. Upper panels: time-lapse images of a representative cell showing Rlc1-GFP localization and mitotic progression monitored by Pcp1-GFP-labeled SPBs (scale bar: 10 µm). The ratios of cells with nodes, mature rings, and constricted rings over time after release from the G2 arrest (as mean ± SD from three different experiments) are shown. Lower panels: Western blot analysis of Rlc1-GFP mobility by Phos-tag SDS-PAGE after release from the G2 block. The image corresponds to a representative experiment that was repeated at least three times with similar results. (**C**) Upper: times for ring assembly and contraction, node condensation/ring maturation, and ring constriction and disassembly were estimated for the indicated strains growing exponentially in YES-glucose (glc) medium, by time-lapse confocal fluorescence microscopy. Mitotic progression was monitored using Pcp1-GFP-labeled SPBs. Lower: ring constriction rates (µm/min), were determined for the indicated strains. n is the total number of cells scored from three independent experiments, and data are presented as violin plots. Statistical comparison between the two groups was performed by unpaired Student's *t*-test. ***, p<0.005; **, p<0.005; *, p<0.05; ns, not significant. (**D**) Decimal dilutions of strains of the indicated genotypes were spotted on solid plates with YES-glucose (glc), or YES-glycerol (gly), incubated at 30 °C for 3 (glc) or 5 days (gly), and photographed. The image corresponds to a representative experiment that was repeated at least three times with similar results. (**E**) Representative maximum projection confocal images of cells grown in YES-glycerol for 12 hr after cell-wall staining with calcofluor white. Scale bar: 10 µm (**F**) Upper: times for total ring assembly and contraction, node condensation/ring maturation, and ring constriction were estimated cells of the indicated strains grown exponentially in YES-glycerol medium by time-lapse fluorescence confocal microscopy. Lower: ring constriction rates (µm/min), were determined for the indicated strains. n is the total number of cells, and data are presented as violin plots. Statistical comparison between the two groups was performed by unpaired Student's *t*-test. ****, p<0.0001; ***, p<0.0005; ns, not significant. (**G**) Representative maximum-projection time-lapse images of Rlc1 dynamics at the equatorial region of cells growing in YES-glycerol. Mitotic progression was monitored using Pcp1-GFP-labeled SPBs. Time interval is 5 min. Scale bar: 10 µm.

The online version of this article includes the following source data and figure supplement(s) for figure 1:

**Source data 1.** Source data for *Figure 1*.

**Source data 2.** Western blot images for *Figure 1A and B*.

**Figure supplement 1.** Rlc1 phosphorylation at Ser35 is essential for *S. pombe* respiratory growth.

**Figure supplement 1—source data 1.** Source data for *Figure 1—figure supplement 1*.

**Figure supplement 1—source data 2.** Western blot images for *Figure 1B*.

**Figure supplement 2.** Myo51 and Ser35-phosphorylated Rlc1 collaborate for contractile actomyosin ring (CAR) assembly during respiration.

**Figure supplement 2—source data 1.** Source data for *Figure 1—figure supplement 2*.

(*Figure 1F–G*). Conversely, expression of the phosphomimetic *rlc1-S35D* allele did not alter CAR dynamics and constriction rate when grown with glycerol (*Figure 1F–G*).

Myo51 is a type V myosin that plays an important role in *S. pombe* during ring assembly, as *myo51Δ* cells complete this process later than normal (*Figure 1—figure supplement 2*; *Laplante et al., 2015*). Myo51 deletion further and specifically increased the CAR assembly time in *rlc1-S35A* cells during glycerol growth (*Figure 1—figure supplement 2*), suggesting that Myo51 cooperates with Myo2 in this process. Thus, in vivo Rlc1 phosphorylation at Ser35 is essential for the modulation of *S. pombe* cytokinesis and cell division during respiratory growth.

## p21-activated kinases Pak2 and Pak1 phosphorylate Rlc1 at Ser35 to positively control fission yeast cytokinesis during respiration

The essential fission yeast p21 (cdc42/rac)-activated protein kinase (PAK) Pak1/Shk1/Orb2, phosphorylates Rlc1 at Ser35 both in vitro and in vivo (*Loo and Balasubramanian, 2008*; *Magliozzi et al., 2020*). Consistently, in vivo phosphorylation of Rlc1-GFP at Ser35 was gradually reduced in glucose-grown cells expressing the analog-sensitive (as) kinase mutant *pak1-M460A* treated with the specific kinase inhibitor 3-BrB-PP1, but not in the presence of the solvent control (*Figure 2—figure supplement 1A*). Interestingly, light chain phosphorylation at Ser35 was absent in glucose-growing cells expressing the hypomorphic mutant allele *pak1-M460G* (*Figure 2A*; *Loo and Balasubramanian, 2008*), suggesting that this kinase version is constitutively inactive towards Rlc1. Unexpectedly, Rlc1 remained phosphorylated at Ser35 in *pak1-M460G* cells during glycerol growth (*Figure 2A*). Rlc1 levels increased by approximately ~1.5-fold during glycerol growth, regardless of the presence of Pak1/2 activity (*Figure 2A*; *Figure 2—figure supplement 1B*), although increased Rlc1 expression does not alter fission yeast CAR integrity and/or cytokinesis (*Stark et al., 2010*). Therefore, other kinase(s) in addition to Pak1 specifically phosphorylate Rlc1 at Ser35 in vivo during respiratory growth.

A reasonable candidate for this role is Pak2, a PAK homolog whose overexpression restores the viability and normal morphology of fission yeast cells in the absence of Pak1 function (*Sells et al.,*

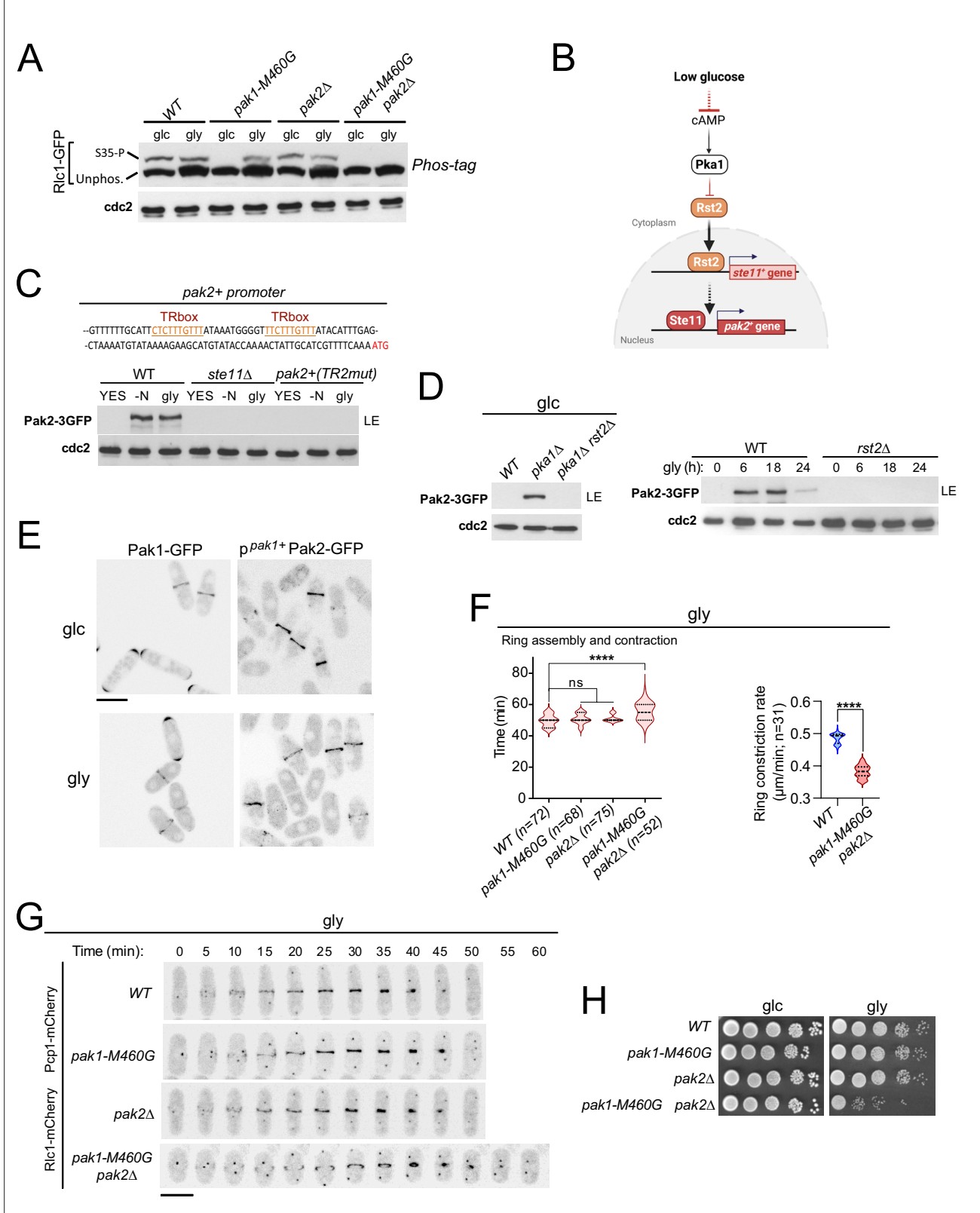

**Figure 2.** p21/Cdc42-activated kinase Pak2 phosphorylates Rlc1 at Ser35 in concert with Pak1 to positively regulate fission yeast cytokinesis during respiratory growth. (**A**) Total protein extracts from strains of the indicated genotypes growing exponentially in YES-glucose (glc) or YES-glycerol (gly), were resolved by Phos-tag SDS-PAGE, and the Rlc1-GFP fusion was detected by incubation with anti-GFP antibody. Anti-Cdc2 was used as a loading control. Rlc1 isoforms, phosphorylated (**S35–P**) and non-phosphorylated at Ser35 (Unphos), are indicated. The blot corresponds to a representative

*Figure 2 continued on next page*

*Figure 2 continued*

experiment that was repeated at least three times and the trend of the mobility shift was reproducible. (**B**) Pak2 expression increases specifically during respiratory growth in an Rst2- and Ste11-dependent manner in the absence of cAMP-PKA signaling. See text for further details. (**C**) Upper: Partial nucleotide sequence of the promoter region of the *pak2+* gene. The two putative Ste11-binding motifs (TR boxes) are shown in color. Lower: Western blot analysis of Pak2-3GFP levels in wild-type, *ste11Δ*, and a mutant strain in which the conserved G in the two putative TR boxes in the *pak2+* promoter was replaced by A, grown in YES-glucose, after nitrogen starvation (-N) and in YES-glycerol (gly) for 12 hr. Pak2-3GFP was detected by incubation with anti-GFP antibody, while anti-Cdc2 was used as a loading control. The image corresponds to a representative experiment, which was repeated at least three times with identical results. (**D**) Left: total protein extracts from strains of the indicated genotypes grown exponentially in YES-glucose (left) or in YES-glycerol for the indicated times (right) were resolved by SDS-PAGE, and the Pak2-3GFP fusion was detected by incubation with anti-GFP antibody. Anti-Cdc2 was used as a loading control. Images are representative of experiments repeated at least three times with identical results. (**E**) Representative maximum projection confocal images of exponentially growing cells from Pak1-GFP and p$^{pak1+}$-Pak2-GFP cells in YES-glucose (glc) or YES-glycerol (gly). (**F**) Total assembly and contraction times (min) and ring constriction rates (μm/min) were estimated by time-lapse confocal fluorescence microscopy for the indicated strains growing exponentially in YES-glycerol (gly) medium. Mitotic progression was monitored using Pcp1-GFP-labeled SPBs. *n* is the total number of cells scored from three independent experiments, and data are presented as violin plots. Statistical comparison between groups was performed by one-way ANOVA. ****, $p<0.0001$; ns, not significant. (**G**) Representative maximum-projection time-lapse images of Rlc1 dynamics at the equatorial region in cells growing with YES-glycerol. Mitotic progression was monitored using Pcp1-GFP-labeled SPBs. The time interval is 5 min. Scale bar: 10 μm. (**H**) Decimal dilutions of strains of the indicated genotypes were spotted on plates with YES-glucose or YES-glycerol, incubated at 30 °C for four days, and photographed. The image corresponds to a representative experiment that was repeated at least three times with similar results.

The online version of this article includes the following source data and figure supplement(s) for figure 2:

**Source data 1.** Source data for *Figure 2*.

**Source data 2.** Western blot images for *Figure 2A, C and D*.

**Figure supplement 1.** Pak2 regulates cytokinesis during respiration.

**Figure supplement 1—source data 1.** Source data for *Figure 2—figure supplement 1*.

**Figure supplement 1—source data 2.** Western blot images for *Figure 1A, B, C and D*.

---

*1998*; *Yang et al., 1998*). Indeed, Rlc1 phosphorylation at Ser35 was absent during respiratory growth in a *pak1-M460G pak2Δ* double mutant, but remained in *pak2Δ* cells grown on either glucose or glycerol (*Figure 2A*). Pak2 was not detected in a glucose-grown strain co-expressing Pak1-GFP and Pak2-3GFP genomic fusions, but its expression level increased in the absence of nitrogen, in the presence of glycerol, or during the stationary phase in a glucose-rich medium (*Figure 2—figure supplement 1C*). Pak2 expression was very low under these conditions, and could only be detected after long exposure times of the immunoblots (>20 min; LE; *Figure 2—figure supplement 1C*). In *S. pombe*, *pak2+* mRNA levels increase during nitrogen starvation through a mechanism dependent on Ste11, a transcription factor that activates gene expression during the early steps of the sexual differentiation ( +) (*Mata and Bähler, 2006*).

The *pak2+* promoter contains two consecutive copies of a putative Ste11-binding motif known as the TR box (consensus sequence 5 +-TTCTTTGTTY-3') (*Figure 2C*; *Sugimoto et al., 1991*). Induced expression of Pak2-3GFP during nitrogen starvation or glycerol growth was abolished in *ste11Δ* cells, and in a strain in which *pak2+* + is controlled by its endogenous promoter mutated at both TR boxes (*Figure 2C*). The Zn-finger transcriptional factor Rst2, whose activity is negatively regulated by the cAMP-PKA signaling pathway in the presence of glucose, positively regulates *ste11+* expression during nitrogen or glucose starvation ( +) (*Kunitomo et al., 2000*). Rst2 deletion suppressed the constitutive expression of Pak2 in glucose-grown *pka1Δ* cells and in the presence of glycerol (*Figure 2D*). Thus, Pak2 expression is constitutively repressed by glucose cAMP-PKA signaling and increases specifically during respiration in an Rst2- and Ste11-dependent manner.

The very low expression levels of the Pak2-3GFP genomic fusion prevented its microscopic visualization during nutrient starvation. To circumvent this situation, we obtained a strain expressing a Pak2-GFP fusion under the control of the native *pak1+* promoter (p$^{pak1+}$-Pak2-GFP). The relative expression levels of p$^{pak1+}$-Pak2-GFP were approximately +–3-fold higher than those of the Pak1-GFP genomic fusion (*Figure 2—figure supplement 1D*). However, in contrast to Pak1-GFP, which is targeted to the cell poles and the CAR during vegetative growth with either glucose or glycerol, the p$^{pak1+}$-Pak2-GFP fusion localized exclusively to the CAR under both conditions (*Figure 2E*). Pak2 co-localized with Rlc1 throughout the cytokinetic process in glycerol-grown cells, from the early steps of

CAR assembly and maturation to the later stages of ring constriction (*Figure 2—figure supplement 1E*).

Compared to wild-type cells, *pak1-M460G* and *pak2Δ* cells showed no defects in cytokinesis, septation, or growth during respiration (*Figure 2F–H*; *Figure 2—figure supplement 1F*). Strikingly, the mean times for CAR assembly and constriction, as well as ring constriction rates, were longer in *pak1-M460G pak2Δ* double mutant cells (*Figure 2F and G*). *pak1-M460G pak2Δ* cells were also multiseptated (*Figure 2—figure supplement 1F*) and displayed a growth defect in this carbon source (*Figure 2H*). Our observations support that Pak1 is entirely responsible for the in vivo Rlc1 phosphorylation at Ser35 during fermentation, whereas Pak2, whose expression is induced upon nutrient starvation, collaborates with Pak1 to phosphorylate Rlc1 at this residue to regulate cytokinesis during respiratory growth.

## Rlc1 phosphorylation is critical for cytokinesis during respiration due to reduced actin cable nucleation imposed by SAPK activation

For3 assembles actin cables for cellular transport and cooperates with the essential formin Cdc12 to nucleate actin filaments for the CAR during cytokinesis (*Coffman et al., 2013*; *Gómez-Gil et al., 2020*). CAR assembly and contraction time increased as the rate of ring constriction was significantly reduced in *for3Δ* cells grown with glycerol (*Figure 3—figure supplement 1A–B*). This led to an accumulation of multiseptated and lysed cells and a marked growth defect (*Figure 3—figure supplement 1C–E*). Thus, For3-mediated actin cable nucleation is crucial for proper cytokinesis and growth of *S. pombe* during respiration.

Glucose limitation activates Sty1, a p38 MAPK ortholog and the key effector of the SAPK pathway in fission yeast (*Madrid et al., 2004*). Activated Sty1 down-regulates CAR integrity in *S. pombe* in response to environmental stress by reducing For3 levels (*Gómez-Gil et al., 2020*). Notably, the transfer of fission yeast cells co-expressing genomic Sty1-HA and For3-3GFP fusions from glucose to glycerol media induced a sustained Sty1 activation and a decrease in For3 protein levels (*Figure 3A*). Similar to environmental stress (*Gómez-Gil et al., 2020*), For3 downregulation is associated with increased ubiquitination, as it was attenuated in the proteasome mutant *mts3-1* (*Figure 3—figure supplement 1F*). The ratio of actin cables to patches was significantly lower in wild-type cells grown in glycerol than in glucose, as revealed by image segmentation analysis of AlexaFluor-488-phalloidin-immunostained cells using the machine learning routine Ilastik (*Figure 3B*; *Berg et al., 2019*). Actin patches appeared partially depolarized and their density increased during glycerol growth (*Figure 3B*). Fluorescence intensity at the cell poles and CAR of activated Cdc42 GTPase (CRIB-3GFP probe), which triggers For3 activation in vivo (*Martin et al., 2007*), decreased during respiration (*Figure 3—figure supplement 2A–B*), as did the targeting of a For3-3GFP fusion (*Figure 3—figure supplement 2C*).

In contrast to wild-type cells (*Figure 3A*), total levels of the constitutively active For3 allele *for3-DAD* fused to GFP, which lacks the intramolecular interaction between the autoregulatory (DAD) and inhibitory (DID) domains to adopt an open and constitutively active conformation (*Martin et al., 2007*), were not reduced upon Sty1 activation by glycerol (*Figure 3C*). *for3-DAD* cells displayed thickened actin cables with an increased actin cable-to-patch ratio (*Figure 3—figure supplement 2D*) and required a shorter time for CAR assembly than the wild-type (*Figure 3D–E*). Most importantly, *for3-DAD* expression largely suppressed the altered cable organization (*Figure 3—figure supplement 2D*), the cytokinetic delay and reduced constriction rate (*Figure 3D–E*), the multiseptated phenotype (*Figure 3F*; *Figure 3—figure supplement 2E*), and the defective growth in glycerol of *rlc1-S35A* cells (*Figure 3G*). *for3-DAD* also rescued the cytokinetic and growth defects in glycerol of *pak1-M460G pak2Δ* cells (*Figure 3D–G*; *Figure 3—figure supplement 2E*), which lack Rlc1 phosphorylation at Ser35 (*Figure 2A*). Nevertheless, CAR assembly and contraction took longer in *for3-DAD rlc1-S35A* cells than in *for3-DAD pak1-M460G pak2Δ* cells (*Figure 3D*), suggesting that the regulation of other targets in addition to Rlc1 by Pak1/2 could affect cytokinesis in fission yeast. Thus, For3 and PAK-phosphorylated Rlc1 play a collaborative and biologically significant role during cytokinesis when *S. pombe* grows by respiration.

In agreement with previous observations (*Gómez-Gil et al., 2020*), For3 levels increased in glucose-grown wild-type or *rlc1-S35A* cells lacking Sty1 activity by deletion of Wis1, the only MAPKK of the SAPK (*Pérez and Cansado, 2010*), and also in the presence of glycerol (*Figure 4A–B*). We could not analyze the cytokinetic and growth defects of *rlc1-S35A* cells during respiration in the absence of Sty1,

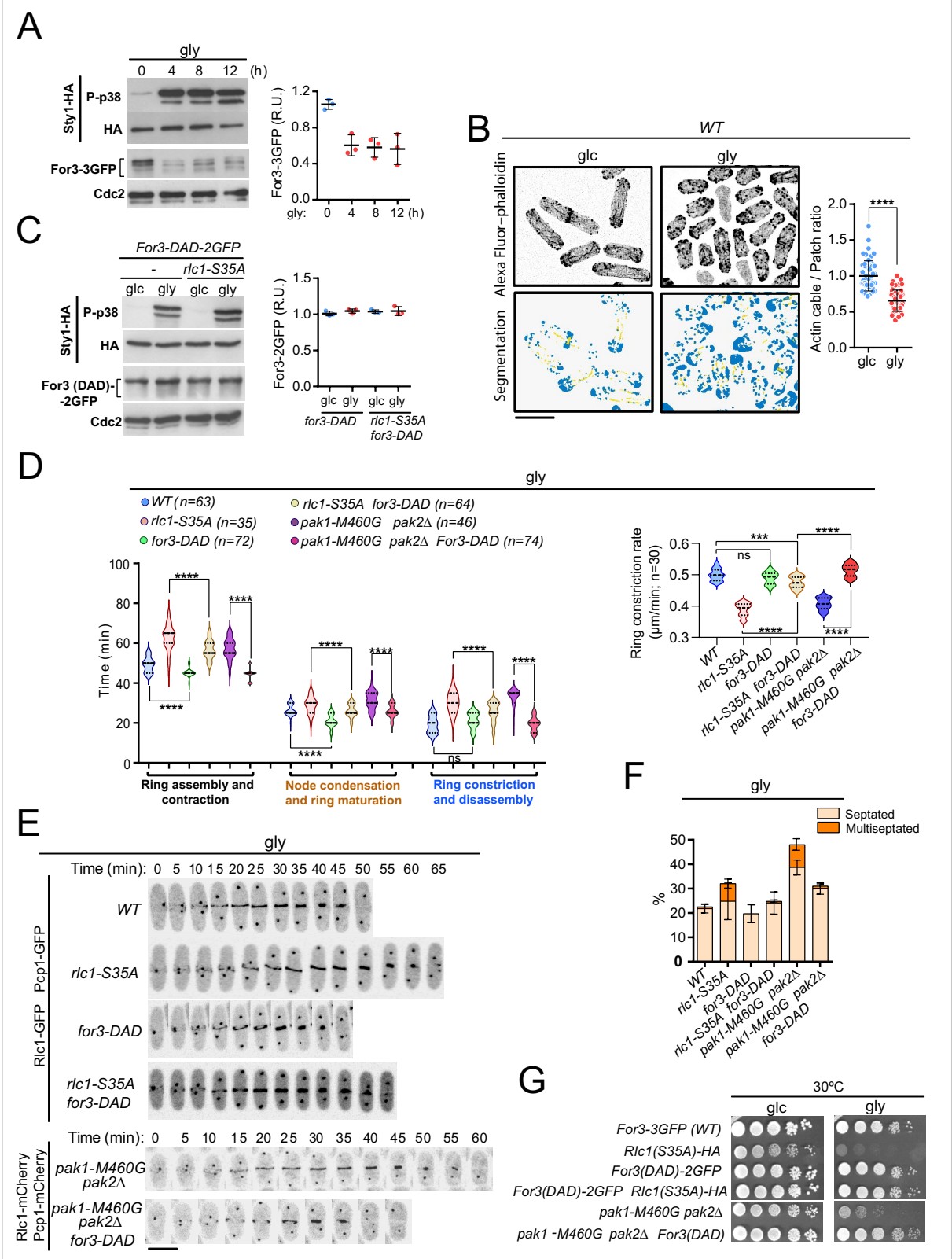

**Figure 3.** PAK phosphorylation of Rlc1 is critical for *S. pombe* cytokinesis during respiration due to impaired For3-dependent actin cable nucleation imposed by SAPK activation. (**A**) Left: *S. pombe* cells expressing genomic Sty1-HA and For3-3GFP fusions were transferred to YES-glycerol medium (gly) for the indicated times. Activated/total Sty1 was detected using anti-phospho-p38 and anti-HA antibodies, respectively. Total For3 was detected with anti-GFP antibody. Anti-Cdc2 was used as a loading control. Right: For3 expression levels are expressed as mean relative units ± SD and correspond to

*Figure 3 continued on next page*

*Figure 3 continued*

experiments performed as biological triplicates. (**B**) Representative maximum projection images of Alexa Fluor phalloidin-stained *S. pombe* cells grown in YES-glucose medium (glc) or in YES-glycerol (gly) for 12 hr. Segmentation analysis using the Ilastik routine is shown below each image. Quantification data correspond to actin cable to patch ratio of G2 cells (n=51) and are presented as mean relative units ± SD. ****, p<0.0001, as calculated by unpaired Student's *t*-test. Scale bar: 10 µm. (**C**) Left: *S. pombe* wild-type and *rlc1-S35A* strains expressing genomic Sty1-HA and For3 (DAD)–2GFP fusions were grown in YES-glucose (glc) until mid-log phase and transferred to YES-glycerol (gly) medium for 12 hr. Activated/total Sty1 was detected using anti-phospho-p38 and anti-HA antibodies, respectively. Total For3 was detected with anti-GFP antibody. Anti-Cdc2 was used as a loading control. Right: For3 expression levels are expressed as mean relative units ± SD and correspond to experiments performed as biological triplicates. (**D**) Left: times for ring assembly and contraction, node condensation/ring maturation, and ring constriction and disassembly were estimated for the indicated strains growing exponentially in YES-glycerol medium by time-lapse confocal fluorescence microscopy. Right: ring constriction rates (µm/min), were determined for the indicated strains. *n* is the total number of cells scored from three independent experiments, and data are presented as violin plots. ****, p<0.0001; ***, p<0.001; ns, not significant, as calculated by unpaired Student's *t*-test. (**E**) Representative maximum-projection time-lapse images of Rlc1 dynamics at the equatorial region in cells from the indicated strains growing in YES-glycerol. Mitotic progression was monitored using Pcp1-GFP-labeled SPBs. Time interval is 5 min. (**F**) The indicated strains were grown in YES-glycerol liquid medium for 12 hr, and the percentage of septated and multiseptated cells was quantified. Data correspond to three independent experiments and are presented as mean ± SD. (**G**) Decimal dilutions of strains of the indicated genotypes were spotted on plates with YES-glucose (glc) or YES-glycerol (gly), incubated at 30 °C or five days, and photographed.

The online version of this article includes the following source data and figure supplement(s) for figure 3:

**Source data 1.** Source data for *Figure 3*.

**Source data 2.** Western blot images for *Figure 3A and C*.

**Figure supplement 1.** For3 formin is required for *S. pombe* cytokinesis during respiratory growth.

**Figure supplement 1—source data 1.** Source data for *Figure 3—figure supplement 1*.

**Figure supplement 1—source data 2.** Western blot images for *Figure 1F*.

**Figure supplement 2.** Localization at the cell poles and the contractile actomyosin ring (CAR) of Ccd42 (active) and For3 is reduced during respiratory growth.

**Figure supplement 2—source data 1.** Source data for *Figure 3—figure supplement 2*.

because, unlike mutants lacking Wis1 or Wak1/Win1 (MAPKKs), *sty1Δ* and *atf1Δ* cells (lacking the Sty1 downstream transcription factor Atf1) are growth defective on this carbon source (*Figure 4—figure supplement 1C*; *Zuin et al., 2008*). Nevertheless, similar to the *sty1Δ* mutant (*Gómez-Gil et al., 2020*), *wis1Δ* cells exhibited thickened actin cables and an increased actin cable-to-patch ratio during growth on glycerol (*Figure 4—figure supplement 1A*). Furthermore, Wis1 deletion increased the actin cable- to-patch ratio in *rlc1-S35A* cells (*Figure 4—figure supplement 1A*), suppressed their delayed cytokinesis and reduced constriction rate (*Figure 4C–D*) and multiseptation (*Figure 4—figure supplement 1B*), and restored cell growth in glycerol (*Figure 4E*).

Lack of Rlc1 phosphorylation at Ser35 has a limited effect on *S. pombe* CAR dynamics during glucose fermentation (*Figure 1C*), where basal Sty1 activity is very low (*Figure 4B*). Our previous data suggest that a constitutive increase in Sty1 activity should be detrimental to cytokinesis in glucose-grown *rlc1-S35A*. Indeed, deletion of the MAPK tyrosine phosphatase Pyp1 (*Figure 4A*), which increases basal Sty1 activity and reduces For3 levels (*Gómez-Gil et al., 2020*), further delayed CAR assembly and contraction time andthe constriction rate of glucose-grown *rlc1-S35A* cells (*Figure 4F–G*), leading to an accumulation of septated cells (*Figure 4—figure supplement 1D*). Therefore, when For3 levels are reduced by SAPK activation, the tight control of Rlc1 function by phosphorylation at Ser35 becomes essential for *S. pombe* cytokinesis.

SAPK-For3 may act downstream of the PAK-Rlc1 cascade, as the cytokinesis and growth defects of *rlc1-S35A* and *pak1-M460G pak2Δ* mutants are suppressed by *for3-DAD*. However, the normal degradation of For3 during respiration in *rlc1-S35A* cells (*Figure 4A*) excludes this possibility. More-over, *for3-DAD* expression did not reduce Rlc1 phosphorylation at Ser35 in the presence of glucose or glycerol (*Figure 4—figure supplement 1E*), suggesting that Pak1/2-Rlc1 is not activated by changes in For3 levels. Thus, both the SAPK-For3 and PAK-Rlc1 pathways act independently and are critical for the successful completion of cytokinesis during respiration.

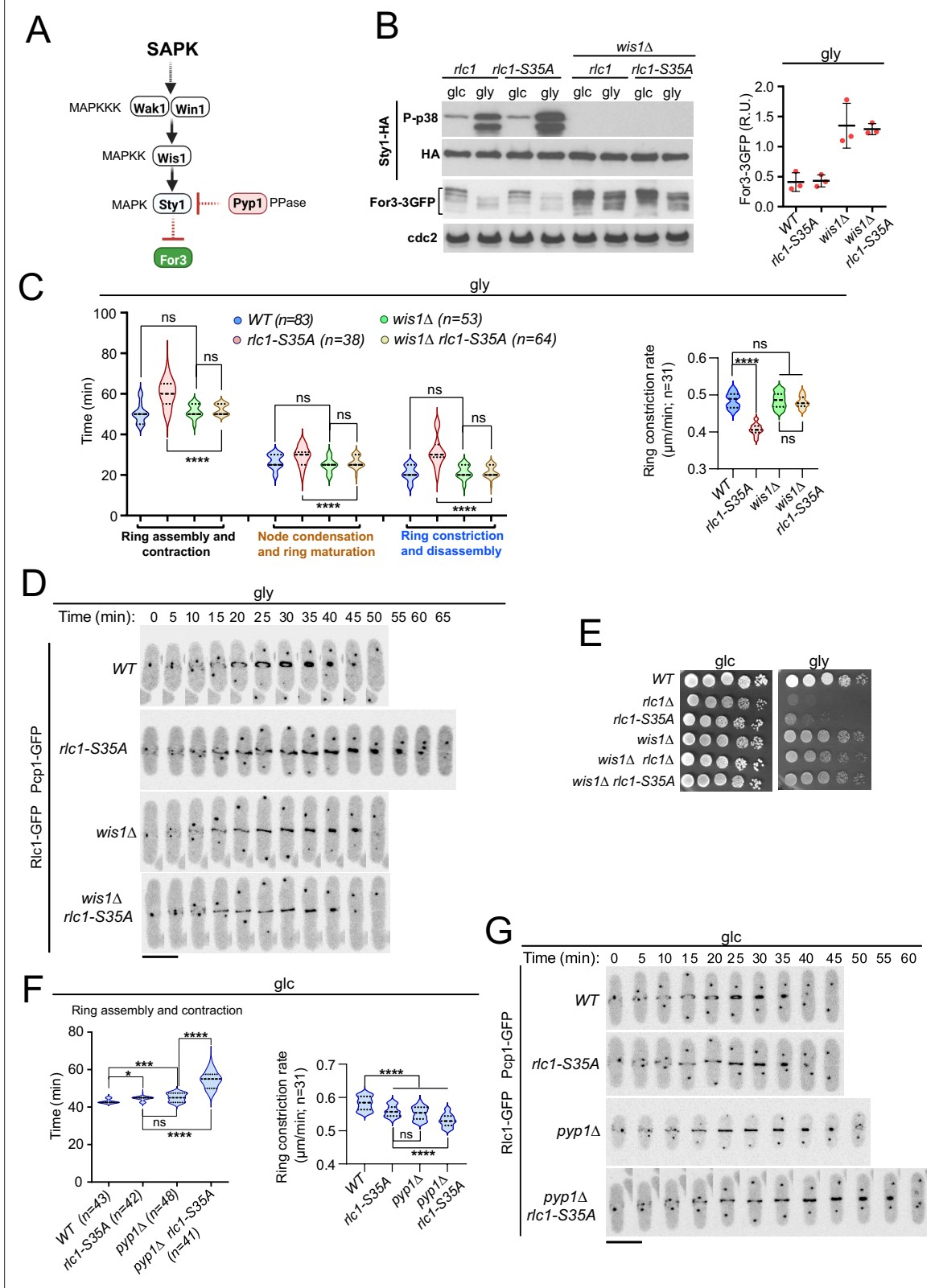

**Figure 4.** Lack of stress-activated protein kinase (SAPK) signaling restores *S. pombe* cytokinesis and growth during respiration in the absence of Rlc1 phosphorylation. (**A**) SAPK signaling triggers For3 downregulation. Please see the text for specific details on the components of the MAPK cascade. (**B**) Left: *S. pombe* strains of the indicated genotypes expressing genomic Sty1-HA and For3-3GFP fusions were grown in either YES-glucose (glc) or YES-glycerol (gly) medium for 12 hr. Activated/total Sty1 were detected with anti-phospho-p38 and anti-HA antibodies, respectively. Total For3 levels

*Figure 4 continued on next page*

*Figure 4 continued*

were detected with anti-GFP antibody. Anti-Cdc2 was used as a loading control. Right: For3 expression levels in glycerol are presented as mean relative units ± SD and correspond to experiments performed as biological triplicates. (**C**) Left: times for ring assembly and contraction, node condensation/ring maturation, and ring constriction and disassembly were estimated for the indicated strains growing exponentially in YES-glycerol medium by time-lapse confocal fluorescence microscopy. Right: ring constriction rates (μm/min), were determined for the indicated strains. *n* is the total number of cells scored from three independent experiments, and data are presented as violin plots. \*\*\*\*, p<0.0001; ns, not significant, as calculated by unpaired Student's *t*-test. (**D**) Representative maximum-projection time-lapse images of Rlc1 dynamics at the equatorial region in cells from the indicated strains growing in YES-glycerol. Mitotic progression was monitored using Pcp1-GFP-labeled SPBs. Time interval is 5 min. (**E**) Decimal dilutions of strains of the indicated genotypes were spotted on plates with YES-glucose (glc) or YES-glycerol (gly), incubated at 30 °C or five days, and photographed. The image corresponds to a representative experiment that was repeated at least three times with similar results. (**F**) Total ring assembly and contraction time (left) and ring constriction rates (μm/min) (right) were determined by time-lapse confocal fluorescence microscopy for the indicated strains growing exponentially in YES-glucose medium (glc). *n* is the total number of cells scored from three independent experiments, and data are presented as violin plots. \*\*\*\*, p<0.0001;\*\*\*, p<0.005; \*, p<0.05; ns, not significant, as calculated by unpaired Student's *t*-test. (**G**) Representative maximum-projection time-lapse images of Rlc1 dynamics at the equatorial region in cells from the indicated strains growing in YES-glucose. Mitotic progression was monitored using Pcp1-GFP-labeled SPBs. Time interval is 5 min.

The online version of this article includes the following source data and figure supplement(s) for figure 4:

**Source data 1.** Source data for *Figure 4*.

**Source data 2.** Western blot images for *Figure 4B*.

**Figure supplement 1.** stress-activated protein kinase (SAPK) activation impairs *S.pombe* cytokinesis and growth during respiration in the absence of Rlc1 phosphorylation.

**Figure supplement 1—source data 1.** Source data for *Figure 4—figure supplement 1*.

**Figure supplement 1—source data 2.** Western blot images for *Figure 1E*.

## Regulation of Myo2 by Rlc1 phosphorylation is essential for *S. pombe* cytokinesis during respiration

The thermosensitive myosin II allele *myo2-E1* shows reduced ATPase activity and actin-filament binding in vitro (*Wang et al., 2020*). Similar to *rlc1-S35A* cells, *myo2-E1* cells showed a growth defect on glycerol. This was not the case for a hypomorphic mutant in the essential myosin II light chain (*cdc4-8*) or for cells lacking the heavy chain Myp2, which cooperates with Myo2 for cytokinesis (*Wang et al., 2020*; *Okada et al., 2019*; *Alonso-Matilla et al., 2019*; *Figure 5A*). Accordingly, CAR assembly and constriction times were much longer in *myo2-E1* cells incubated at a semi-restrictive temperature (30 °C), and switched to the permissive temperature in a glycerol-based medium compared to glucose (78.59±11.09 *vs* 61.67±7.56 min, respectively) (*Figure 5B–C*). The respiration-induced cytokinetic delay was more pronounced during ring constriction/disassembly, with a much slower constriction rate (*Figure 5B–C*) and the accumulation of multiseptated cells with thickened septa (*Figure 5—figure supplement 1A*). However, the delay in CAR closure was very similar between *myp2Δ* and wild-type cells were grown with glucose or glycerol (~7.5 *vs*~7.7 min) (*Figure 5—figure supplement 2A*). The *myp2Δ* mutant accumulated multiseptated cells in the presence of glycerol, and this phenotype was exacerbated when combined with the *rlc1-S35A* allele (*Figure 5—figure supplement 2B*).

Expression of *for3-DAD* in *myo2-E1* cells restored their altered cable organization (*Figure 5—figure supplement 1B*), suppressed the cytokinetic delay, the slow rate of ring constriction, and the multiseptated phenotype (*Figure 5D–E*; *Figure 5—figure supplement 1C*), and defective growth with glycerol at semi-restrictive temperatures (*Figure 5F*). Therefore, regulation of Myo2 function by in vivo phosphorylated Rlc1 is critical for *S. pombe* cytokinesis and division during respiration due to the reduced actin filament nucleation imposed by SAPK activation.

## Exogenous antioxidants bypass the need to regulate Myo2 by Rlc1 phosphorylation during respiratory growth cytokinesis

Animal cells produce reactive oxygen species (ROS) during aerobic respiration due to electron leakage from mitochondria (*Ostrow et al., 1994*). Oxidative stress from free radicals produced during respiration in *S. pombe* (*Malina et al., 2021*) activates Sty1 and antioxidant responses at both transcriptional and translational levels (*Figure 6A*; *Zambon et al., 2017*). Remarkably, 0.16 mM of the antioxidant tripeptide reduced glutathione (GSH) in the growth medium counteracted many effects of oxidative stress in *rlc1-S35A* and *myo2-E1* cells grown in glycerol. Changes included recovery of For3 levels

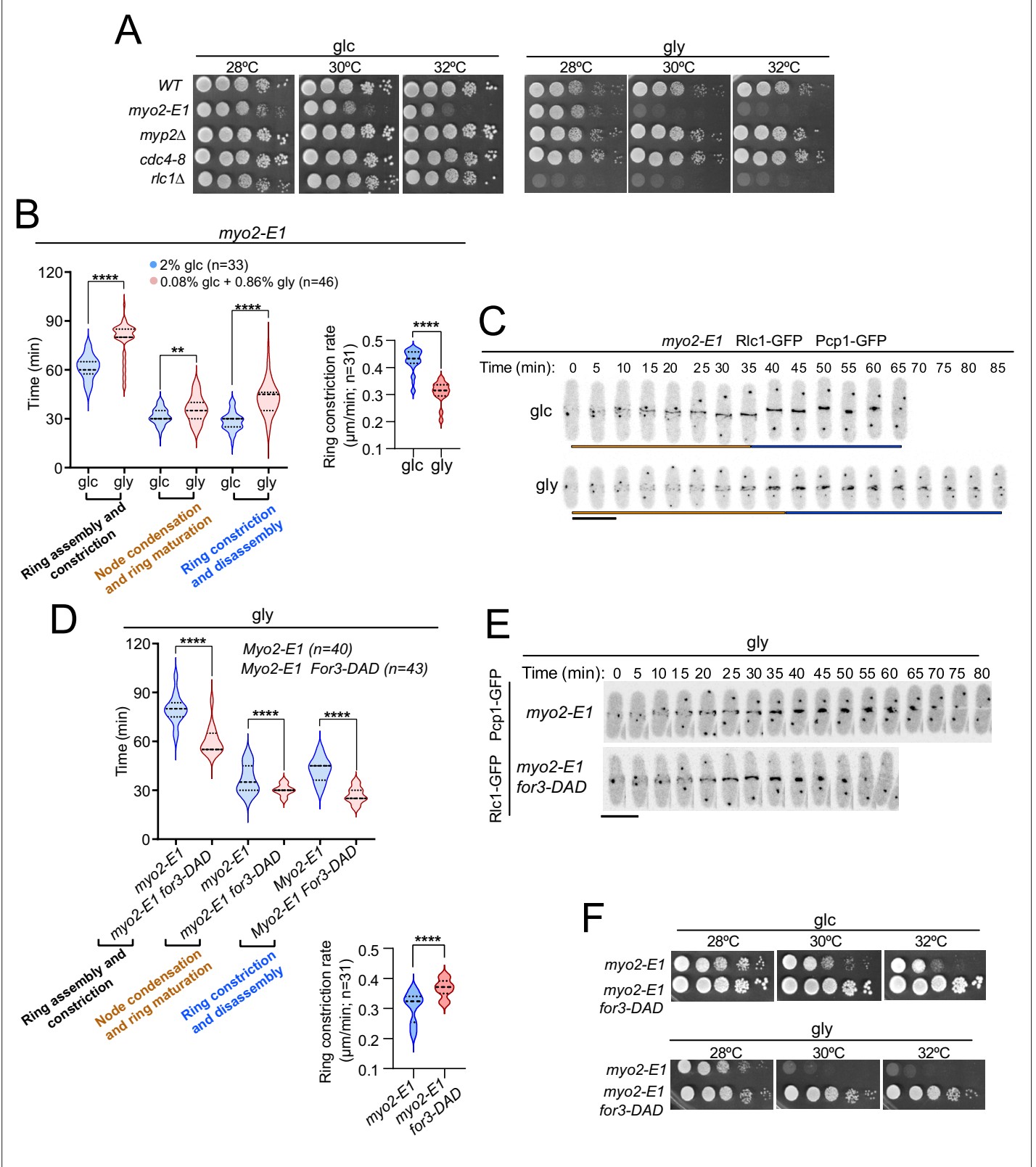

**Figure 5.** Control of Myo2 activity by Rlc1 phosphorylation regulates *S. pombe* cytokinesis and growth during respiration. (**A**) Decimal dilutions of strains of the indicated genotypes were spotted on plates with YES-glucose or YES-glycerol, incubated at 28, 30, and 32 °C for three (glc) or five (gly) days, and photographed. The images correspond to a representative experiment, which was repeated at least three times with similar results. (**B**) Left: times for ring assembly and contraction, node condensation/ring maturation, and ring constriction and disassembly were estimated for *myo2-E1* cells growing in

*Figure 5 continued on next page*

*Figure 5 continued*

YES-glucose (glc) and YES-glycerol medium (gly) by time-lapse confocal fluorescence microscopy. Mitotic progression was monitored using Pcp1-GFP-labeled SPBs. Right: ring constriction rates (µm/min). *n* is the total number of cells scored from three independent experiments, and data are presented as violin plots. ****, p<0.0001; **, p<0.005, as calculated by unpaired Student's *t*-test. (**C**) Representative maximum-projection time-lapse images of Rlc1 dynamics at the equatorial region in *myo2-E1* Rlc1-GFP cells growing in YES-glucose (glc) or YES-glycerol (gly). Mitotic progression was monitored using Pcp1-GFP-marked SPBs. Time interval is 5 min. Scale bar: 10 µm. (**D**) Upper: times for ring assembly and contraction, node condensation/ring maturation, and ring constriction and disassembly were estimated for the indicated strains grown on YES-glycerol (gly) using time-lapse confocal fluorescence microscopy. Lower: ring constriction rates (µm/min). *n* is the total number of cells scored from three independent experiments, and data are presented as violin plots. ****, p<0.0001, as calculated by unpaired Student's *t*-test. (**E**) Representative maximum-projection time-lapse images of Rlc1-GFP dynamics at the equatorial region in *myo2-E1* and *myo2-E1 for3-DAD* cells growing in YES-glycerol. Mitotic progression was monitored using Pcp1-GFP-labeled SPBs. Time interval is 5 min. (**F**) Decimal dilutions of strains of the indicated genotypes were spotted on plates with YES-glucose or YES-glycerol, incubated at 28, 30, and 32°C for three (glc) or five (gly) days, and photographed. The images correspond to a representative experiment that was repeated at least three times with similar results.

The online version of this article includes the following source data and figure supplement(s) for figure 5:

**Source data 1.** Source data for *Figure 5*.

**Figure supplement 1.** *for3-DAD* expression restores actin filaments and alleviates the septation defects of *myo2-E1* cells during respiration.

**Figure supplement 1—source data 1.** Source data for *Figure 5—figure supplement 1*.

**Figure supplement 2.** Role of Myp2 on *S. pombe* cytokinesis during respiration.

**Figure supplement 2—source data 1.** Source data for *Figure 5—figure supplement 2*.

---

(*Figure 6A*), the cable-to-patch ratio (*Figure 6—figure supplement 1A*), the timing of CAR assembly/contraction and constriction rate (*Figure 6B–C*), septation (*Figure 6—figure supplement 1B*) and growth (*Figure 6D*). Normal growth depended on For3 (*Figure 6D*). In contrast, in *wis1Δ* cells, GSH had no significant effect on the above-mentioned characteristics (*Figure 6B–D*). Hence, respiration-induced oxidative stress reduces the nucleation of actin cables by formins and renders the execution of cytokinesis dependent on the phosphorylation of the Myo2 light chain Rlc1.

## Discussion

RLC phosphorylation regulates myosin II activity in both muscle and non-muscle cells. It plays a key positive role as a regulator of myosin II function in cardiac muscle contraction under normal and disease conditions (*Yuan et al., 2015*). In non-muscle vertebrate cells, RLC phosphorylation at Ser19 is essential for NMII contractile activity during cell migration and division (*Garrido-Casado et al., 2021*; *Komatsu et al., 2000*). In *Drosophila melanogaster*, in vivo phosphorylation of *spaghetti-squash* RLC at the conserved Ser21 is critical for myosin II activation, preventing embryonic lethality and severe cytokinesis defects (*Jordan and Karess, 1997*). Conversely, in the unicellular amoeba *Dictyostelium discoideum* RLC phosphorylation at the conserved Ser13 is not essential for myosin II function (*Ostrow et al., 1994*).

The role of RLC phosphorylation on myosin II activity during cytokinesis in the fission yeast *S. pombe* has remained elusive. While some studies suggest that lack of Rlc1 phosphorylation at the conserved Ser35 delays CAR constriction (*Pollard et al., 2017*; *Loo and Balasubramanian, 2008*), others indicate the opposite (*Sladewski et al., 2009*). *S. pombe* uses aerobic fermentation instead of respiration for ATP production when glucose is available, while mitochondrial energy metabolism is significantly reduced (*Malina et al., 2021*). To our knowledge, all published studies investigating the mechanistic insights of fission yeast cytokinesis have been performed in glucose-fermenting cells. Here, we show that Rlc1 phosphorylation plays a modest role during cytokinesis in the presence of glucose, but becomes essential when *S. pombe* switches to respiratory metabolism. In this metabolic state, lack of Rlc1 phosphorylation at Ser35 results in a significant delay in CAR assembly and constriction, leading to multiseptation and limited growth using the respiratory carbon source glycerol. To allow cytokinesis during respiration, Ser35-phosphorylated Rlc1 targets Myo2, the major myosin II heavy chain involved in fission yeast CAR assembly and constriction (*Zambon et al., 2017*). Accordingly, the cytokinetic defects of cells expressing the hypomorphic allele *myo2-E1*, which exhibits reduced ATPase activity and actin-filament binding (*Wang et al., 2020*), are exacerbated during respiration and resemble those of *rlc1-S35A* cells. Therefore, respiratory carbohydrate metabolism dictates the biological relevance of Rlc1 phosphorylation in modulating Myo2 activity during *S. pombe* cytokinesis.

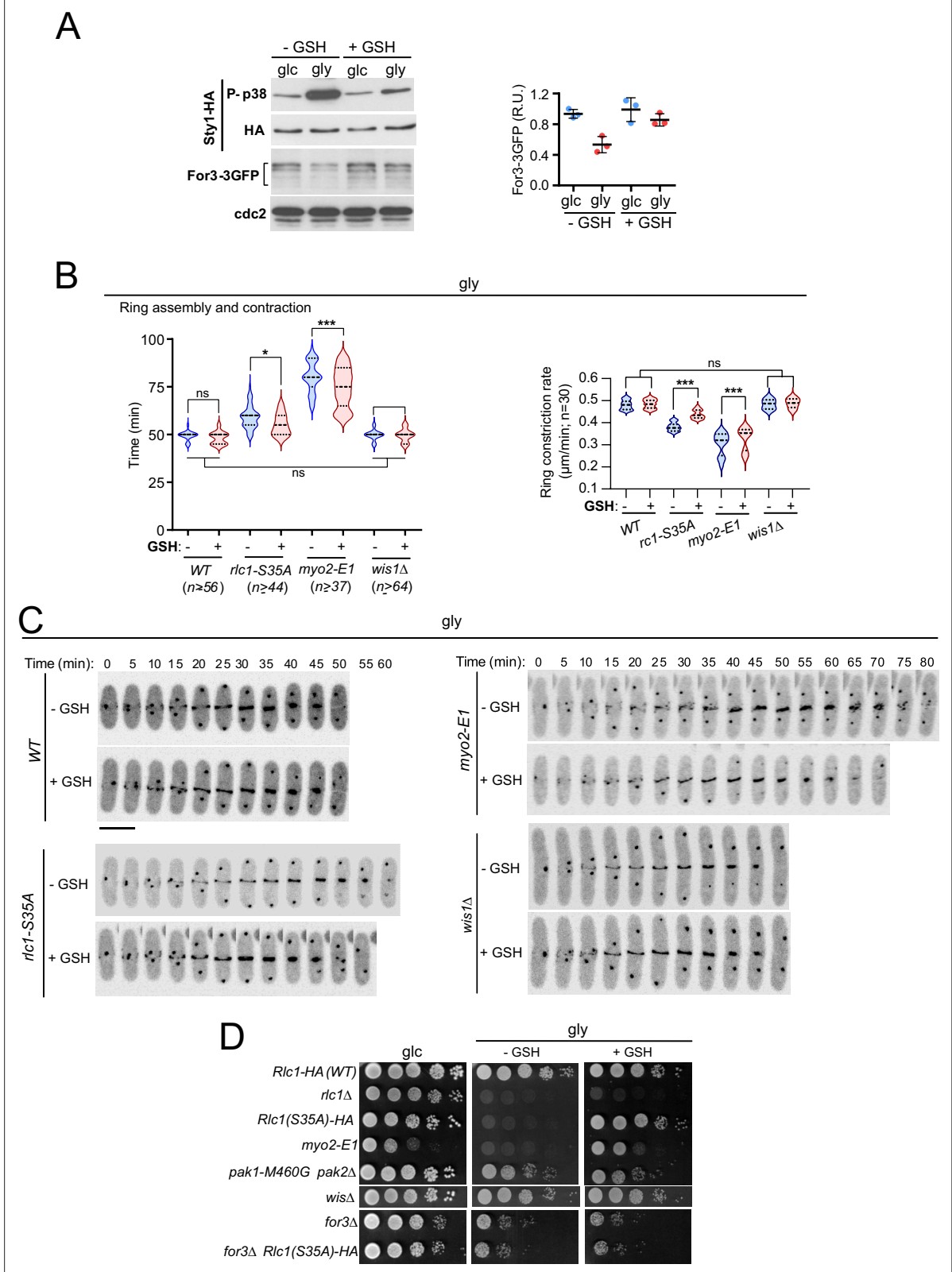

**Figure 6.** Exogenous antioxidants bypass the need for Rlc1 phosphorylation to regulate myosin II activity and cytokinesis during respiratory growth. (**A**) Left: *S. pombe* wild-type cells expressing genomic Sty1-HA and For3-3GFP fusions were grown to mid-log phase in YES-glucose (glc) or YES-glycerol (gly), with or without 0.16 mM reduced glutathione (GSH). Activated/total Sty1 was detected with anti-phospho-p38 and anti-HA antibodies, respectively. Total For3 levels were detected with anti-GFP antibody. Anti-Cdc2 was used as a loading control. Right: For3 expression levels are expressed as mean

*Figure 6 continued on next page*

*Figure 6 continued*

relative units ± SD and correspond to experiments performed as biological triplicates. (**B**) Total ring assembly and contraction times (left) and ring constriction rates (µm/min) (right) were estimated by time-lapse confocal fluorescence microscopy for the indicated strains growing exponentially in YES-glycerol medium with or without 0.16 mM GSH. *n* is the total number of cells scored from three independent experiments, and data are presented as violin plots. ***, p<0.005; *, p<0.05; ns, not significant, as calculated by unpaired Student's *t*-test. (**C**) Representative maximum-projection time-lapse images of Rlc1 dynamics at the equatorial region in cells from the indicated strains growing in YES-glycerol with or without 0.16 mM GSH. Mitotic progression was monitored using Pcp1-GFP-labeled SPBs. Time interval is 5 min. Scale bar: 10 µm. (**D**) Decimal dilutions of strains of the indicated genotypes were spotted onto plates with YES-glucose or YES-glycerol plates with or without 0.16 mM GSH, incubated at 28 °C for three (Glc) or five (Gly) days, and photographed. The images correspond to a representative experiment that was repeated at least three times with similar results.

The online version of this article includes the following source data and figure supplement(s) for figure 6:

**Source data 1.** Source data for *Figure 6*.

**Source data 2.** Western blot images for *Figure 6A*.

**Figure supplement 1.** Reduced glutathione restores actin cables and alleviates septation defects in *rlc1-S35A* and *myo2-E1* cells during respiration.

**Figure supplement 1—source data 1.** Source data for *Figure 6—figure supplement 1*.

**Figure supplement 2.** Model depicting the signaling pathways and mechanisms that regulate *S. pombe* cytokinesis by myosin II (Myo2) through regulatory light chain phosphorylation during respiratory metabolism.

The p21-activated kinase Pak1 phosphorylates Rlc1 at Ser35 in vivo in a glucose-rich medium (*Loo and Balasubramanian, 2008*; *Magliozzi et al., 2020*). However, evidence from this work supports that Pak2 phosphorylates Rlc1 at this residue together with Pak1 for adequate CAR contractility during respiration. Accordingly, cytokinetic and respiratory growth defects similar to those reported for *rlc1-S35A* cells were observed in the absence of both Pak1 and Pak2 activity. Pak2 expression is induced during respiratory growth by a transcriptional mechanism involving the Ste11 transcription factor. In turn, Ste11 expression is activated by the Rst2 transcription factor, whose activity is repressed by the cAMP-PKA pathway in the presence of glucose (*Kunitomo et al., 2000*), making Pak2 available during respiration. In this situation, Pak2 may boost PAK activity by acting on multiple targets, some of them redundantly with Pak1. In addition to Rlc1, phosphoproteomic screens have identified other Pak1 substrates that are functional during cytokinesis and polarized growth (*Magliozzi et al., 2020*). However, since Pak2 only localizes to the CAR during respiratory growth, its functional redundancy with Pak1 might be limited to cytokinesis-associated proteins.

In animal cells, both de novo actin disposal at the division site and cortical actin transport/flow contribute to CAR assembly (*Cao and Wang, 1990*; *Khaliullin et al., 2018*; *White and Borisy, 1983*). In fission yeast cells, which lack an actin filament cortex, the CAR is mainly assembled by Myo2 from actin filaments nucleated de novo at the cytokinesis nodes by the essential formin Cdc12, and partly from Cdc12-nucleating actin cables pulled from the non-equatorial zone (*Huang et al., 2012*; *Pelham and Chang, 2002*). For3, the formin that assembles the actin cables involved in polarized secretion and growth, also contributes to CAR formation in fission yeast (*Coffman et al., 2013*; *Gómez-Gil et al., 2020*). In turn, the activated SAPK pathway down-regulates CAR assembly and stability in response to stress by reducing For3 levels (*Gómez-Gil et al., 2020*; *Madrid et al., 2021*). Like animal cells, fission yeast cells undergo endogenous oxidative stress during respiration (*Farrugia and Balzan, 2012*), which leads to Sty1 activation and downregulation of For3 levels and actin filament availability. Hence, in this metabolic context, the regulation of Myo2 function by Rlc1 phosphorylation during cytokinesis becomes critical due to a decrease in For3-nucleated actin filaments. Accordingly, restoring actin filament availability by various strategies, including the expression of a constitutively active For3 version, limiting For3 downregulation in SAPK-less mutants, or attenuating endogenous oxidative stress with antioxidants (GSH), rescued cytokinesis and cell growth during respiration in both *rlc1-S35A* and *myo2-E1* mutants. The number of actin filaments at the CAR is reduced by approximately half in *myo2-E1* cells (*Malla et al., 2022*). Enhanced nucleation of actin filaments by For3 may therefore alleviate their defective actin-binding and motor activity during cytokinesis.

Metabolic reprogramming drives the actin cytoskeletal rearrangements that occur during cell response to external forces, epithelial-to-mesenchymal transition, and cell migration (*Bays et al., 2017*; *Shiraishi et al., 2015*; *DeWane et al., 2021*). However, an open question is how changes in cell metabolism trigger actin cytoskeleton remodeling (*DeWane et al., 2021*). Our observations reveal a sophisticated adaptive interplay between modulation of myosin II function by Rlc1 phosphorylation

and environmentally controlled formin availability, which becomes critical for a successful cytokinesis during a respiratory carbohydrate metabolism (*Figure 6—figure supplement 2*). Collectively, these findings provide a remarkable example of how carbohydrate metabolism dictates the relative importance of different sources of actin filaments for CAR dynamics during cellular division.

# Materials and methods

**Key resources table**

| Reagent type (species) or resource | Designation | Source or reference | Identifiers | Additional information |
|---|---|---|---|---|
| Antibody | anti-Phospho-p38 (rabbit polyclonal) | Cell Signaling | Cat# 9211, RRID:AB_331641 | WB (1:1000) |
| Antibody | anti-HA (mouse monoclonal) | Roche | Cat# 11 583 816 001, RRID:AB_514505 | WB (1:1000) |
| Antibody | anti-GFP (mouse monoclonal) | Roche | Cat# 11 814 460 001, RRID:AB_390913 | WB (1:1000) |
| Antibody | HRP-conjugated anti-HA antibody (rat monoclonal) | Roche | Cat# 12 013 819 001, RRID:AB_390917 | WB (1:3000) |
| Antibody | anti-Cdk1/Cdc2 (PSTAIR) (rabbit polyclonal) | Millipore | Cat#: 06–923; RRID:AB_310302 | WB (1:1000) |
| Antibody | anti-Mouse IgG- peroxidase (goat polyclonal) | Sigma Aldrich | Cat#: A5278; RRID:AB_258232 | WB (1:2000) |
| Antibody | anti-Rabbit IgG- peroxidase (goat polyclonal) | Sigma Aldrich | Cat#: A6667; RRID:AB_258307 | WB (1:2000) |
| Commercial assay, kit | ECL Western Blotting Reagents | GE-Healthcare | Cat#: RPN2106 | |
| Chemical compound, drug | β-estradiol | Sigma Aldrich | Cat#: E2758 | 10–500 μM |
| Chemical compound, drug | PhosTag acrylamide | Wako Chemical | Cat#: 300–93523 | 15 μM |
| Chemical compound, drug | PP1 Analog III, 3-MB-PP1 | Sigma Aldrich | Cat#: 529582 | 1 μM |
| Chemical compound, drug | Alexa fluor 488-conjugated phalloidin | Thermo Fischer Scientific | Cat#: A12379 | 200 units/ml (~6.6 μM) |
| Chemical compound, drug | Soybean lectin | Sigma Aldrich | Cat#: L2650 | 1 mg/ml |
| Software, algorithm | ImageJ | ImageJ | https://imagej.net/Fiji/Downloads | Quantification of Western blots and microscopic analysis |
| Software, algorithm | Graphpad Prism 9.0.2 | Graphpad | https://www.graphpad.com/scientific-software/prism// | Statistical analysis and graphs representation |
| Software, algorithm | Ilastik | Ilastik | https://www.ilastik.org/ | Segmentation toolkit |
| Other | μ-Slide 8 well | Ibidi | Cat#: 80826 | For time-lapse imaging of CAR dynamics. (Microscopy analysis; Materials and Methods section) |

## Strain construction

*S. pombe* strains used in this work are listed in *Supplementary file 1*. Several deletion strains were obtained from the Bioneer mutant library (*Kim et al., 2010*), while null mutants in the *rlc1⁺*, *pka1⁺*, *ste11⁺*, and *rst2⁺* genes were obtained by ORF deletion and replacement with G418 (kanR), nourseothricin (NAT), or hygromycin B cassettes using a PCR-mediated strategy (*Hentges et al., 2005*; *Sato et al., 2005*) and the oligonucleotides described in . Strains expressing different genomic fusions were constructed either by transformation or by random spore analysis of appropriate crosses in sporulation agar (SPA) medium (*Petersen and Russell, 2016*). To generate a strain expressing an integrated Rlc1-HA fusion, the *rlc1⁺* + plus its endogenous promoter was amplified by PCR using genomic DNA

from *S. pombe* 972h⁻ wild-type strain as a template, and the 5' and 3'oligonucleotides PromRlc1(Xhol)-FWD and Rlc1-HA(SacII)-REV (*Supplementary file 2*), which contain a *Xho*I restriction site and an extended DNA sequence encoding a HA C-terminal tag plus a *Sac*II site, respectively. The *Xho*I-*Sac*II digested PCR fragment was cloned into plasmid pJK210 (*Keeney and Boeke, 1994*), sequenced, linearized with *Bmg*BI, and transformed into an *rlc1Δ ura4.294* strain. To obtain a strain expressing an integrative Rlc1-HA fusion under the control of the β-estradiol promoter (*Ohira et al., 2017*), the *rlc1⁺* ORF fused to a 3 + HA tag was amplified by PCR using the 5' and 3-oligonucleotides Rlc1 (SmaI)-FWD and Rlc1-HA (SacII)-REV, containing *Sma*I and *Sac*II restriction sites, respectively. The amplified PCR product was cloned into a modified plasmid pJK210 containing a β-estradiol regulated promoter Z₃EV (*Ohira et al., 2017*), and the resulting construct was linearized with *Stu*I, and transformed into an *rlc1Δ ura4.294* strain. To obtain a strain expressing an integrative Rlc1-GFP fusion, DNA encoding an Rlc1-GFP fusion under the endogenous promoter was amplified by PCR using as a template genomic DNA from a *S. pombe* strain expressing a genomic Rlc1-GFP fusion (*Supplementary file 1*) and the 5' and 3' oligonucleotides PromRlc1(XhoI)-FWD and Rlc1-GFP(SacII)-REV containing *Sma*I and *Sac*II restriction sites, respectively. In all cases, *ura4⁺* + were obtained, and the correct integration and expression of the Rlc1-HA and Rlc1-GFP fusions under either the endogenous or the β-estradiol regulated promoters were verified by both PCR and Western blot analysis. To generate strains expressing Rlc1-GFP versions with mutations at residues Ser35 and Ser36 to alanine or aspartic acid, the pJK210 plasmid containing the Rlc1-GFP fusion was used as a template for site-directed mutagenesis by PCR, using specific mutagenic oligonucleotides described in *Supplementary file 2*. The mutagenized plasmids were then linearized with *Bmg*BI and transformed into an *rlc1Δ ura4.294* strain.

The *S. pombe* strain expressing a genomic Pak2-3GFP fusion was obtained in two consecutive steps. First, the *pak2⁺* + plus its endogenous promoter was amplified by PCR using genomic DNA from the *S. pombe* 972h⁻ wild-type strain as a template, and the 5' and 3' oligonucleotides PromPak2(XhoI)-FWD (*Xho*I site) and Pak2GFP(SmaI/XmaI)-REV (*Sma*I site) (*Supplementary file 2*). The PCR product was cloned in a frame into a pJK210 plasmid containing a GFP C-terminal tag. In a second step, this construct was linearized with *Sma*I and two additional GFP tags were added by a Gibson assembly approach. Finally, the resulting plasmid was linearized with *Bmg*BI and transformed into a *pak2Δ ura4.294* strain. To introduce the mutations at the two putative Ste11-binding motifs (TR box) in the Pak2 promoter, the pJK210-Pak2-3GFP plasmid was subjected to sequential site-directed mutagenesis by PCR. In this way, the conserved G in each motif was replaced by A by using the mutagenic oligonucleotides described in *Supplementary file 2*. To generate a strain that produces a Pak2-GFP fusion under the control of Pak1 promoter, the Pak1 5'-UTR sequence was amplified by PCR using genomic DNA from the wild-type *S. pombe* 972h⁻ strain, and assembled by Gibson cloning to a PCR-amplified Pak2-GFP fragment and the pJK210 plasmid linearized with *Sma*I. The resulting plasmid was digested with *Bmg*BI and transformed into a *pak2Δ ura4.294* strain.

## Media and growth conditions

In liquid culture experiments, fission yeast strains were grown overnight with shaking at 28 °C in YES-glucose medium containing 0.6% yeast extract, 2% glucose and supplemented with adenine, leucine, histidine, or uracil (100 mg/liter) (*Prieto-Ruiz et al., 2020*). The next day, cultures were diluted to an OD₆₀₀ of 0.01 and incubated to a final OD₆₀₀ of 0.2. Then, cells were recovered by filtration, washed three times, and shifted to either YES-glucose or YES-glycerol (0.6% yeast extract, 0.08% glucose, 0.86% glycerol, plus supplements), and incubated at 28 °C for 4 or 8 hr before imaging. In experiments performed with the *myo2-E1* mutant, cells harvested from cultures at 28 °C were resuspended in YES-glucose or YES-glycerol, incubated at 30 °C for 2 hr, and then at 28 °C for the remainder of the experiment. In experiments with cells expressing the analog-sensitive Cdc2 (CDK) kinase version *cdc2-asM17* (*Aoi et al., 2014*), cells from log-phase liquid cultures in YES-glucose (OD₆₀₀ 0.5), were treated with 1 µM 3-NM-PP1 (Sigma-Aldrich, 529581) for 3.5 hr, recovered by filtration, washed, and resuspended in YES-glucose medium. In experiments with strains expressing an analog-sensitive Pak1 kinase version *pak1-M460A*, log-phase liquid cultures were split in two and incubated for different times in YES-glucose medium treated with 10 µM 3-BrB-PP1 (Abcam, ab143756), or in medium lacking the analog kinase inhibitor. For nitrogen starvation experiments, strains growing exponentially in Edinburgh Minimal Medium (EMM2) (*Moreno et al., 1991*) containing 2% glucose (OD₆₀₀ 0.5), were recovered by filtration and resuspended in the same medium without ammonium chloride for the indicated

times. For the plate assays of growth stress sensitivity, *S. pombe* wild-type and mutant strains were grown in YES-glucose liquid medium to an $OD_{600}$ of 1.2, recovered by centrifugation, resuspended in YES to a density of $10^7$ cells/ml, and appropriate decimal dilutions were spotted on YES-glucose, or YES-glycerol solid plates (2% agar). The plates were incubated for 3 days (YES-glucose) or 5 days (YES-glycerol), at different temperatures (28 °C, 30 °C, 32 °C, and/or 34 °C), depending on the experiment, and then photographed. All the assays were repeated at least three times with similar results. Representative experiments are shown in the corresponding figures. When required, solid and/or liquid media were supplemented with varying amounts of β-estradiol (Sigma-Aldrich, RPN2106) or reduced glutathione (GSH; Sigma-Aldrich, G6013).

## Microscopy analysis

For *time-lapse* imaging of CAR dynamics, 300 µl of cells grown exponentially for 4 hr in YES-glucose or YES-glycerol liquid medium, and prepared as described above, were added to one well of a µ-Slide eight well chamber (Ibidi, 80826) previously coated with 10 µl of 1 mg/ml soybean lectin (Sigma-Aldrich, L2650) (*Gómez-Gil et al., 2020*). GSH was added to the medium as required, to at a final concentration of 0.3 mM. Cells were allowed to sediment in the culture media and adhere to the bottom of the well for 1 min, and images were taken every 2.5 min for 2 hr in YES-glucose cultures or every 5 min for 8 hr in YES-glycerol cultures. Experiments were performed at 28 °C, and single mid-planes were taken from a set of six stacks (0.61 µm each) at the indicated time points. Time-lapse images were acquired using a Leica Stellaris eight confocal microscope with a 63 X/1.40 Plan Apo objective controlled by the LAS X software. The time for node condensation and ring maturation includes the time from SPB separation to the onset of CR constriction. Ring constriction and disassembly time include the time from the first frame of ring constriction to the last frame where the ring is completely constricted and disassembled. The total time for ring assembly and contraction is the sum of these two values. *n* is the total number of cells scored from at least three independent experiments. Ring constriction rates were measured manually as the average circumference ± SD of wild-type and mutant Rlc1-GFP versions at contractile rings in each time frame starting from the first frame of CAR constriction. n=30 cells were measured for each strain and growth condition. Statistical comparison between the two groups was performed by unpaired Student's *t-test*.

For actin staining with Alexa-Fluor phalloidin, 5 ml mid-log cultures were grown in YES-glucose ($OD_{600}$ 0.5) or YES-glycerol ($OD_{600}$ 0.2) for 12 hr after the media shift. Cells were fixed by immersion for 1 h in 3.7% formaldehyde in PEM buffer (10 mM EGTA; 1 mM $MgSO_4$; 100 mM PIPES pH 6.9, 75 mM sucrose, and 0.1% Triton X-100). After three washes with PEM, the cell pellets were resuspended in 20 µl of cold 40% methanol solution, stained with 8 µl of 5 mg/ml Alexa Fluor 488-conjugated phalloidin (Thermo Fisher Scientific, A12379), and incubated overnight at 4 °C in the dark on a rotary platform. Images of stained cells were captured from samples spotted on glass slides using a Leica Stellaris eight confocal microscope with a 100 X/1.40 Plan Apo objective (seven stacks of 0.3 µm each). For actin segmentation analysis, the Ilastik routine using the pixel classification tool (*Berg et al., 2019*) was trained with two representative images, one of the cells growing in YES-glucose medium and one of the cells growing in YES-glycerol. Training involved drawing cables, patches, and backgrounds in three different colors. Once the program was trained, the remaining images from the different experiments were uploaded to Ilastik to perform the segmentation routine. The resulting images were then exported to ImageJ (*Schneider et al., 2012*) and the segmented cells at G2 were analyzed using the color histogram tool to obtain the specific areas corresponding to cables and patches. Data from n≥40 cells growing in YES-glycerol were obtained for each cell by dividing the cable area by the patch area, and the ratio was normalized with respect to the average obtained from wild-type cells growing in YES-glucose medium. For For3-GFP quantification, Ilastik was trained by drawing For3-GFP dots, GFP background, and image background in three different colors. The ratio of For3-GFP spots to cytosol was calculated by dividing the area of For3-GFP spots by the area of GFP background of at least n≥40 cells in G2 or late M (dividing cells) and normalized to the average of the glucose ratio.

For cell wall staining, cells were recovered from 1 ml aliquots by centrifugation, stained with 1 µl of 0.5 mg/ml calcofluor white, and images were acquired from samples spotted on glass slides using a Leica Stellaris 8 confocal microscope with a 63 X/1.40 Plan Apo objective (six stacks of 0.61 µm each). The percentage of septated (one septa), multiseptated (two or more septa), and lysed cells was

calculated at the indicated time points for each strain and condition from three independent experiments. n≥100 cells were counted from multiple images taken during each replicate.

## Western blot analysis

To determine the level of Rlc1-GFP fusion and/or its phosphorylation status, 10 ml samples of fission yeast cultures were collected and precipitated with TCA (*Grallert and Hagan, 2017*). Protein extracts were resolved on 12% SDS-PAGE gels containing 30 µM Phos-tag acrylamide (Wako, AAL-107), transferred to nitrocellulose blotting membranes, and immunoblotted with a mouse monoclonal anti-GFP antibody (Roche, 11 814 460 001, RRID:AB_390913). Rabbit monoclonal anti-PSTAIR (anti-Cdc2; Sigma-Aldrich, 06–923, RRID:AB_310302) was used as a loading control. Immunoreactive bands were detected using anti-mouse (Abcam, ab205719, RRID:AB_2755049), and anti-rabbit HRP-conjugated secondary antibodies (Abcam, ab205718, RRID:AB_2819160), and the ECL system (GE-Healthcare, RPN2106). For the detection of Pak1-GFP, Pak2-3GFP, For3-3GFP, and For3(DAD)–2GFP fusions, protein extracts obtained after TCA-precipitation (Pak1-GFP and Pak2-3GFP fusions) or under native conditions (For3-3GFP and For3(DAD)–2GFP fusions) were resolved on 6% SDS-PAGE gels, transferred to Hybond-ECL membranes, and incubated with a mouse monoclonal anti-GFP antibody (Roche) and anti-cdc2 (PSTAIR) as a loading control. In all cases, the immunoreactive bands were revealed using anti-mouse or anti-rabbit HRP-conjugated secondary antibodies and the ECL system. Detection of Sty1 phosphorylation and total protein levels in strains expressing a genomic Sty1-HA fusion was performed exactly as described in *Gómez-Gil et al., 2020*. Dual phosphorylation of Sty1 was detected using a rabbit polyclonal anti-phospho-p38 antibody (Cell Signaling, 9211, RRID:AB_331641). Total Sty1 was detected in *S. pombe* extracts using a mouse monoclonal anti-HA antibody (12CA5, Roche). Immunoreactive bands were detected using anti-mouse or anti-rabbit HRP-conjugated secondary antibodies (Abcam) and the ECL system.

Densitometric quantification of Western blot experiments from 16-bit. jpg digital images of blots were performed using ImageJ (*Schneider et al., 2012*). The bands of interest plus background were drawn as rectangles and a profile plot (peak) was generated for each band. To reduce background noise in the bands, each peak above the baseline was manually closed using the straight-line tool. Measurement of the closed peaks was performed using the wand tool. Relative units (R.U.) of For3 levels were estimated by determining the signal ratio of the corresponding anti-GFP (total For3) blot with respect to the anti-cdc2 blot (internal control) at each time point. Quantification data correspond to experiments performed as biological triplicates. Mean relative units ± SD is shown.

## Statistical analysis

Statistical analysis was performed using prism 9.0.2. software (GraphPad), and results are represented as violin plots or mean ± SD, unless otherwise stated. Comparisons between two groups were calculated using unpaired two-tailed Student's t-tests, whereas comparisons between more than two groups were calculated using one-way ANOVA with Bonferroni's multiple comparison test. We observed normal distribution and no difference in variance between groups in individual comparisons. Statistical significance: *$p < 0.05$; **$p < 0.005$; ***$p < 0.0005$; ****$p < 0.0001$. Further details of statistical analysis are given in the figure legends.

## Acknowledgements

We thank Pedro M Coll for providing yeast strains and plasmids. This research was funded by Agencia Estatal de Investigación and Ministerio de Ciencia e Innovación, Spain, grant numbers PID2020-112569GB-I00 and PGC2018-098924-B-I00, and the Regional Government of Castile and Leon, Spain, grant number CSI150P20. European Regional Development Fund (ERDF), co-funding from the European Union. FP-R and AP-D are, respectively, Formación de Profesorado Universitario Ph.D. fellows from Ministerio de Educación y Formación Profesional and Universidad de Murcia, Spain.

## Additional information

### Funding

| Funder | Grant reference number | Author |
| --- | --- | --- |
| Agencia Estatal de Investigación | PID2020-112569GB-I00 | José Cansado |
| Agencia Estatal de Investigación | PGC2018-098924-B-I00 | Pilar Pérez |
| Regional Government of Castile and Leon | CSI150P20 | Pilar Pérez |

The funders had no role in study design, data collection and interpretation, or the decision to submit the work for publication.

### Author contributions

Francisco Prieto-Ruiz, Data curation, Software, Formal analysis, Investigation, Visualization, Methodology, Writing – review and editing; Elisa Gómez-Gil, Resources, Formal analysis, Validation, Investigation, Methodology; Rebeca Martín-García, Resources, Data curation, Formal analysis, Investigation, Methodology; Armando Jesús Pérez-Díaz, Alejandro Franco, Formal analysis, Investigation, Methodology; Jero Vicente-Soler, Resources, Formal analysis, Validation, Investigation, Methodology, Project administration, Writing – review and editing; Teresa Soto, Data curation, Validation, Investigation, Methodology; Pilar Pérez, Conceptualization, Resources, Supervision, Funding acquisition, Validation, Visualization, Writing – review and editing; Marisa Madrid, Conceptualization, Data curation, Supervision, Validation, Investigation, Visualization, Writing – review and editing; José Cansado, Conceptualization, Supervision, Funding acquisition, Validation, Writing – original draft, Writing – review and editing

### Author ORCIDs

Armando Jesús Pérez-Díaz http://orcid.org/0000-0002-5494-0087
Jero Vicente-Soler http://orcid.org/0000-0001-8759-6545
Alejandro Franco http://orcid.org/0000-0001-7461-3414
Teresa Soto http://orcid.org/0000-0003-2965-318X
José Cansado http://orcid.org/0000-0002-2342-8152

### Decision letter and Author response

Decision letter https://doi.org/10.7554/eLife.83285.sa1
Author response https://doi.org/10.7554/eLife.83285.sa2

## Additional files

### Supplementary files
- MDAR checklist
- Supplementary file 1. S.pombe strains used in this study.
- Supplementary file 2. Oligonucleotides and DNA fragments used in this study.

### Data availability

All data generated or analysed during this study are included in the manuscript and supporting files.

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
