## [Editor Report]

This research advance manuscript breaks new ground by linking cytokinesis regulation to myosin light chain phosphorylation that is dependent on whether cells are growing through respiration or fermentation. This is an exciting new direction and will spur more activity in uncovering adaptations/modifications of cytokinesis mechanisms to metabolic states.

---

## [Decision Letter]

**Decision letter after peer review:**

Thank you for submitting your article "Myosin II regulatory light chain phosphorylation and formin availability modulate cytokinesis upon changes in carbohydrate metabolism" for consideration by *eLife*. Your article has been reviewed by 3 peer reviewers, and the evaluation has been overseen by a Reviewing Editor (Mohan Balasubramanian) and Anna Akhmanova as the Senior Editor. The following individual involved in the review of your submission has agreed to reveal their identity: Thomas D Pollard (Reviewer #2).

The reviewers have discussed their reviews with one another, and the Reviewing Editor has drafted this to help you prepare a revised submission. The referees have all commented that the paper needs a very thorough rewriting to make every section more succinct and less speculative. Please revise according to all the points raised by the three referees. The revisions will make this a much stronger paper.

In addition, three experiments are required, listed below.

The referees' comments are pasted below verbatim. Please use their suggestions for the improvement of the manuscript.

Essential experimental revisions:

(1) Use available phosphomimetic Rlc1 alleles (e.g., Sladewski et al., MBoC 2009) to provide orthogonal evidence for the authors' conclusions.

(2) As far as I could tell, all data concerning the interplay between the two pathways (SAPK-For3 and PAK-Rlc1) are based on the rescue of the latter by expression of For3-DAD. In a simple scheme, this would put SAPK-For3 downstream of PAK-Rlc1; however, normal degradation of For3 in Rlc1-S35A cells (Figure 4A) rejects this model. I believe this point deserves to be emphasized in the manuscript. This leaves two other models: two independent pathways (Figure S6), or PAK1-Rlc1 being activated in response to reduced For3. These can be dissected by testing if For3-DAD expression suppresses Rlc1 phosphorylation.

(3) I strongly suggest that the effects of ROS and antioxidants in sty1∆ be examined.

*Reviewer #1 (Recommendations for the authors):*

(A) CAR assembly and contraction analysis

The phenotypic demonstration as "CAR assembly and disassembly" should be changed to "CAR assembly and contraction." Disassembly may give an impression of corruption in CAR.

Moreover, these two events should be analyzed individually, the ring assembly time and ring contraction rate.

Like Figure 1C/G/H and Figure 5B/C, the authors should break down all the phenotypic analyses of the CAR to the (1) ring assembly time and (2) ring contraction rate, especially in Figures 3 and 4.

(B) Schematic representation

The authors showed a model as a supplemental figure.

Readers should appreciate this to understand better the relationships between SAPK, Git pathway, Ste11/Rst2 pathway, Cdc42 pathway, and Rlc1 phosphorylation. However, this manuscript does not discuss how phospho-Rlc1 facilitates CAR assembly and contraction. Therefore, it is better not to include schematics of F-actin and myosin head interactions. Indeed, under the respiratory metabolism state, the model showed less interaction of the myosin head with F-actin, which is not discussed in this manuscript. Could the author simplify the figures to hide myosin and F-actin contacts?

*Reviewer #2 (Recommendations for the authors):*

I edited one paragraph starting on line 316 as an example of how the text could be much more concise. Editing as recommended here could reduce the size of the text by almost half and make it more reader-friendly.

The authors wrote: "In animal cells, aerobic respiration is accompanied by the production of reactive oxygen species (ROS), which typically arise due to electron leakage from the mitochondrial electron transport chain [49]. *S. pombe* cells also produce free radicals during respiratory growth [50], and the ensuing endogenous oxidative stress prompts Sty1 activation and an antioxidant response both at transcriptional and translational levels [51] (Figure 6A). Remarkably, the increased Sty1 basal activity of wild-type and rlc1- S35A cells growing with glycerol was largely counteracted in the presence of 0.16 mM of reduced glutathione (GSH), an antioxidant tripeptide, and this was accompanied by a recover in total For3 levels (Figure 6A). As expected, segmentation analysis confirmed that the cable-to-patch ratio was significantly improved in glycerol-growing rlc1-S35A and Myo2-E1 cells supplemented with GSH, compared to those growing without the antioxidant (Figure 6B-C). Furthermore, the delayed CAR constriction and disassembly, and the multiseptation of both rlc1-S35A or Myo2-E1 cells during growth with glycerol, were alleviated in the presence of GSH (Figure 6D-G). Most importantly, the simple addition of GSH allowed rlc1-S35A and Myo2-E1 cells to resume growth and proliferate in the presence of glycerol (Figure 6H). This growth-recovery phenotype was For3-dependent since it was not shown by rlc1-S35A for3Δ cells incubated in the presence of the antioxidant (Figure 6H). Hence, oxidative stress is the leading cause of the formin-dependent reduction in the nucleation of actin cables, and imposes regulation of Myo2 function by Rlc1 phosphorylation as a critical factor in the execution of cytokinesis during respiration." (249 words)

Edited: "Animal cells produce reactive oxygen species (ROS) during aerobic respiration due to electron leakage from mitochondria [49]. Oxidative stress from free radicals in *S. pombe* [50] activates Sty1 and antioxidant responses in both transcription and translation [51] (Figure 6A). (This paragraph is the background.)

Remarkably, 0.16 mM of the antioxidant tripeptide reduced glutathione (GSH) in the growth medium counteracted many effects of oxidative stress in rlc1-S35A and myo2-E1 cells grown in glycerol. Changes included recovery of total For3 levels (Figure 6A), the cable-to-patch ratio (Figure 6B-C), the timing of CAR constriction, disassembly and septation (Figure 6D-G) and growth (Figure 6H). Normal growth depended on For3 (Figure 6H). Hence, during respiration oxidative stress reduces nucleation of actin cables by formins and makes the execution of cytokinesis dependent on phosphorylation of Rlc1the Myo2 light chain." (128 words for two paragraphs) (This paragraph presents the new data.)

Line 72: "Myp2 plays a subtle non-essential role for cytokinesis during unperturbed growth." Note that Ref 14 shows that Myp2 contributes as much as Myo2 to contractile ring constriction.

Line 81: Consider a separate paragraph for the sentence "Remarkably, in response…"

Line 96: What is the meaning of myo2 motility in "Myo2 motility is reduced in fission yeast cells expressing an rlc1-S35A S36A mutant version."

Lines 124 and 141. Start new paragraphs.

Line 158: "The cytokinetic delay was mainly observed during the stage of ring constriction and disassembly." Might Myo5 be covering for Myo2 during ring assembly as shown in Ref 14?

Line 163: extra "with" at the end of the line.

Line 186: "relative expression of Pak2 was always very low compared to Pak1 and could only be detected after long exposure times of immunoblots." How do you know that the low signal is due to low numbers of molecules or an insensitive antibody?

Line 215: I would rephrase for clarity: "did not display defects in cytokinesis, septation or growth." Also, "the average times for CAR assembly and disassembly were longer in the Pak1-M460G pak2Δ double mutant cells."

Line 257: "engrossed" is a poor choice of words. Engrossed means something else than intended.

Line 259: Is for3-DAD the same as For3(DAD)? A short phrase in the main text explaining the mutated protein structure would avoid having to look up the reference.

Line 322: reword "this was accompanied by recovery of total For3 levels."

Line 873: I do not understand what you mean by "ring assembly and disassembly" vs. "ring constriction and disassembly." Why is disassembly included with assembly? Figure 3F has the same problem.

Throughout: myosin, myo2 (gene), glucose and glycerol are not proper nouns.

*Reviewer #3 (Recommendations for the authors):*

To address the points of weakness, the authors may consider the following.

(1) Use available phosphomimetic Rlc1 alleles (e.g., Sladewski et al., MBoC 2009) to provide orthogonal evidence for the authors' conclusions.

(2) As far as I could tell, all data concerning the interplay between the two pathways (SAPK-For3 and PAK-Rlc1) are based on the rescue of the latter by expression of For3-DAD. In a simple scheme, this would put SAPK-For3 downstream of PAK-Rlc1; however, normal degradation of For3 in Rlc1-S35A cells (Figure 4A) rejects this model. I believe this point deserves to be emphasized in the manuscript. This leaves two other models: two independent pathways (Figure S6), or PAK1-Rlc1 being activated in response to reduced For3. These can be dissected by testing if For3-DAD expression suppresses Rlc1 phosphorylation.

(3) As stated in the above section, I strongly suggest that the effects of ROS and antioxidants in sty1∆ be examined.

---

## [Author Response]

Reviewer #1 (Recommendations for the authors):(A) CAR assembly and contraction analysisThe phenotypic demonstration as "CAR assembly and disassembly" should be changed to "CAR assembly and contraction." Disassembly may give an impression of corruption in CAR.

Following the reviewer´s advice, the terms “*CAR assembly and disassembly"* have been changed to *"CAR assembly and contraction"* throughout the revised manuscript, including the Figures and Figure legends.

Moreover, these two events should be analyzed individually, the ring assembly time and ring contraction rate.Like Figure 1C/G/H and Figure 5B/C, the authors should break down all the phenotypic analyses of the CAR to the (1) ring assembly time and (2) ring contraction rate, especially in Figures 3 and 4.

The ring assembly times and ring contraction rates have been determined as suggested, and are included in all the modified Figures 1 to 6 (plus supplementary figures) in the revised manuscript. Text describing these new data has been included when necessary in several places in the Results section within the revised version.

(B) Schematic representationThe authors showed a model as a supplemental figure.Readers should appreciate this to understand better the relationships between SAPK, Git pathway, Ste11/Rst2 pathway, Cdc42 pathway, and Rlc1 phosphorylation. However, this manuscript does not discuss how phospho-Rlc1 facilitates CAR assembly and contraction. Therefore, it is better not to include schematics of F-actin and myosin head interactions. Indeed, under the respiratory metabolism state, the model showed less interaction of the myosin head with F-actin, which is not discussed in this manuscript. Could the author simplify the figures to hide myosin and F-actin contacts?

The modified model in the revised manuscript still shows the basic node organization of Myo2 and the F-actin included in the original Figure, but with one important change. The space between the myosin heads and the actin cables has been kept identical both during fermentation and respiration, and includes 3 question marks (?). This presentation avoids any possible misunderstanding about the putative mechanism by which Rlc1 phosphorylation affects Myo2 actin affinity and adtivity during CAR assembly and contraction, which, as the reviewer notes, is not addressed in this work. Furthermore, in the modified model, both For3 and F-actin appear “blurred” during respiration due to For3 downregulation induced by Sty1 MAPK activation. This suggests that, under these specific conditions, the limited availability of actin cables imposes a strict control on Myo2 activity by phosphorylated Rlc1, which is required for an optimal modulation of cytokinesis.

Reviewer #2 (Recommendations for the authors):I edited one paragraph starting on line 316 as an example of how the text could be much more concise. Editing as recommended here could reduce the size of the text by almost half and make it more reader-friendly.The authors wrote: "In animal cells, aerobic respiration is accompanied by the production of reactive oxygen species (ROS), which typically arise due to electron leakage from the mitochondrial electron transport chain [49]. *S. pombe* cells also produce free radicals during respiratory growth [50], and the ensuing endogenous oxidative stress prompts Sty1 activation and an antioxidant response both at transcriptional and translational levels [51] (Figure 6A). Remarkably, the increased Sty1 basal activity of wild-type and rlc1- S35A cells growing with glycerol was largely counteracted in the presence of 0.16 mM of reduced glutathione (GSH), an antioxidant tripeptide, and this was accompanied by a recover in total For3 levels (Figure 6A). As expected, segmentation analysis confirmed that the cable-to-patch ratio was significantly improved in glycerol-growing rlc1-S35A and Myo2-E1 cells supplemented with GSH, compared to those growing without the antioxidant (Figure 6B-C). Furthermore, the delayed CAR constriction and disassembly, and the multiseptation of both rlc1-S35A or Myo2-E1 cells during growth with glycerol, were alleviated in the presence of GSH (Figure 6D-G). Most importantly, the simple addition of GSH allowed rlc1-S35A and Myo2-E1 cells to resume growth and proliferate in the presence of glycerol (Figure 6H). This growth-recovery phenotype was For3-dependent since it was not shown by rlc1-S35A for3Δ cells incubated in the presence of the antioxidant (Figure 6H). Hence, oxidative stress is the leading cause of the formin-dependent reduction in the nucleation of actin cables, and imposes regulation of Myo2 function by Rlc1 phosphorylation as a critical factor in the execution of cytokinesis during respiration." (249 words)Edited: "Animal cells produce reactive oxygen species (ROS) during aerobic respiration due to electron leakage from mitochondria [49]. Oxidative stress from free radicals in *S. pombe* [50] activates Sty1 and antioxidant responses in both transcription and translation [51] (Figure 6A). (This paragraph is the background.)Remarkably, 0.16 mM of the antioxidant tripeptide reduced glutathione (GSH) in the growth medium counteracted many effects of oxidative stress in rlc1-S35A and myo2-E1 cells grown in glycerol. Changes included recovery of total For3 levels (Figure 6A), the cable-to-patch ratio (Figure 6B-C), the timing of CAR constriction, disassembly and septation (Figure 6D-G) and growth (Figure 6H). Normal growth depended on For3 (Figure 6H). Hence, during respiration oxidative stress reduces nucleation of actin cables by formins and makes the execution of cytokinesis dependent on phosphorylation of Rlc1the Myo2 light chain." (128 words for two paragraphs) (This paragraph presents the new data.)

We thank the reviewer for editing and reducing the size of the text in the above paragraph, which is included in the revised manuscript. We have also made significant efforts to reduce the text of the manuscript in each section, with more compact phrasing (including the headlines for the different Results sections), and more short paragraphs to make the paper more readable. This has resulted in an overall reduction in the total number of words in the manuscript from ~11.000 to 9.000 (including Abstract, Introduction, Results, Discussion, Materials and methods, and Figure legends sections), equivalent to approximately four pages of typed text.

Line 72: "Myp2 plays a subtle non-essential role for cytokinesis during unperturbed growth." Note that Ref 14 shows that Myp2 contributes as much as Myo2 to contractile ring constriction.

Agree. The sentence has been changed in the revised manuscript to read:

“Myo2 is essential for viability and cytokinesis during unperturbed growth, whereas Myp2 plays a non-essential but important role during CAR constriction, and in response to salt stress [14-16].”

Line 81: Consider a separate paragraph for the sentence "Remarkably, in response…"

The modified version now includes a separate paragraph, as suggested by the reviewer.

Line 96: What is the meaning of myo2 motility in "Myo2 motility is reduced in fission yeast cells expressing an rlc1-S35A S36A mutant version."

This refers to a previous work by Matthew Lord’s group (Reference number 31: Sladewski TE, *et al.* Mol Biol Cell. 2009;20(17):3941-52), which showed that the average rate of in vitro motility for purified Myo2 bound to the Rlc1-*S35A S36A* mutant is reduced by ∼25% compared to that of Myo2 bound to Rlc1-*S35D S36D* or wild-type Rlc1. As the reviewer notes, this is not clearly described in the original manuscript, so new text has been added in the revised version to read:

“However, another study described that the average in vitro motility rate of purified Myo2 bound to the Rlc1-*S35A S36A* mutant is reduced by ∼25% compared to that of the myosin bound to wild-type Rlc1, and that phosphorylation at both sites is positive for CAR constriction dynamics [31].”

Lines 124 and 141. Start new paragraphs.

Two new paragraphs have been started in the revised manuscript as recommended.

Line 158: "The cytokinetic delay was mainly observed during the stage of ring constriction and disassembly." Might Myo5 be covering for Myo2 during ring assembly as shown in Ref 14?

In response to the possibility raised by the reviewer, we have investigated this. As shown in the new Figure 1—figure supplement 2 in the revised manuscript, we found that the delayed CAR assembly of *myo51Δ* cells was aggravated in the presence of an *rlc1-S35A* allele during growth with glycerol, suggesting that Myo51 cooperates with Myo2 in this process as proposed.

The corresponding Results section in the revised manuscript now reads:

“Myo51 is a type V myosin that plays an important role in *S. pombe* during ring assembly, as *myo51∆* cells complete this process later than normal (Figure 1—figure supplement 2) [14]. Myo51 deletion further and specifically increased the CAR assembly time in *rlc1-S35A* cells during glycerol growth (Figure 1—figure supplement 2), suggesting that Myo51 cooperates with Myo2 in this process.”

Line 163: extra "with" at the end of the line.

The extra “with” has been eliminated in the revised version.

Line 186: "relative expression of Pak2 was always very low compared to Pak1 and could only be detected after long exposure times of immunoblots." How do you know that the low signal is due to low numbers of molecules or an insensitive antibody?

Both possibilities are in our opinion highly unlikely since the Pak2-GFP fusion can be easily detected when expressed in glycerol-growing cells under the control of the Pak1 promoter instead of its own promoter (please see Figure 2—figure supplement 1D in the revised manuscript). Thus, despite being de-repressed, Pak2 expression is low during respiration.

Line 215: I would rephrase for clarity: "did not display defects in cytokinesis, septation or growth." Also, "the average times for CAR assembly and disassembly were longer in the Pak1-M460G pak2Δ double mutant cells."

Both sentences have been rephrased in the revised manuscript as suggested.

Line 257: "engrossed" is a poor choice of words. Engrossed means something else than intended.

This is truly a poor choice of words. The term has been changed to “*thickened*” in the revised manuscript.

Line 259: Is for3-DAD the same as For3(DAD)?

Yes, it is the same allele, whereas For3(DAD)-2GFP refers to the protein fusion.

The *for3-DAD* allele lacks the intra-molecular interaction between the autoregulatory (DAD) and inhibitory (DID) domains, and is therefore constitutively active (*Martin SG,et al.. Mol Biol Cell. 2007;18(10):4155-67*; reference 43 in the revised manuscript). To avoid confusion, *for3-DAD* has been used in the revised text.

A short phrase in the main text explaining the mutated protein structure would avoid having to look up the reference.

Text describing the main mutations in the constitutively active for3-DAD allele has been included in the revised manuscript as follows:

“In contrast to wild-type cells (Figure 3A), total levels of the constitutively active For3 allele *for3-DAD* fused to GFP, which lacks the intramolecular interaction between the autoregulatory (DAD) and inhibitory (DID) domains to adopt an open and constitutively active conformation [43], were not reduced upon Sty1 activation by glycerol (Figure 3C).”

Line 322: reword "this was accompanied by recovery of total For3 levels."

This line has been rewritten with the whole paragraph in the revised manuscript by following the reviewer´s advice in Major point 1 above.

Line 873: I do not understand what you mean by "ring assembly and disassembly" vs. "ring constriction and disassembly." Why is disassembly included with assembly? Figure 3F has the same problem.

Taking into account the reviewer’s opinion, which is also shared by reviewer #1 (please see major point A above), the terms “*CAR assembly and disassembly"* have been changed to *"CAR assembly and contraction"* throughout the revised manuscript, including all the Figures and the corresponding Figure legends.

Throughout: myosin, myo2 (gene), glucose and glycerol are not proper nouns.

OK. They are now written in lowercase throughout the revised manuscript, including figures and figure legends.

Reviewer #3 (Recommendations for the authors):To address the points of weakness, the authors may consider the following.(1) Use available phosphomimetic Rlc1 alleles (e.g., Sladewski et al., MBoC 2009) to provide orthogonal evidence for the authors' conclusions.

Following the reviewer’s suggestion, we have performed a series of new experiments with cells expressing a phosphomimetic a *rlc1-S35D* allele, as shown in Figure 1 of the original manuscript. These new experiments include quantifying CAR dynamics and ring constriction rates during growth on glucose (fermentation) and glycerol (respiration). As can be seen in the revised Figure 1C, and confirming previous observations with the *rlc1-S35D S36D* allele by Sladewsky *et al.*, CAR dynamics and ring constriction rates were identical between wild-type and *rlc1-S35D* cells. Similarly, the phosphomimetic *rlc1-S35D* allele did not alter CAR dynamics and constriction rate during glycerol-growth, which were identical to those of wild-type cells (Figures 1D, 1F and 1G).

We have included explanatory text on this issue in several places within the Results section of the revised manuscript:

“Cells expressing a dual phospho-mimicking form of Rlc1 (Rlc1-S35D S36D) exhibit normal CAR dynamics and support cytokinesis like wild-type cells [31]. Similarly, CAR dynamics and ring constriction rates were identical between wild-type and Rlc1(S35D)-GFP cells (Figure 1C).”

“Strikingly, the unphosphorylated mutants *rlc1-S35A* and *rlc1-S35A S36A*, but not *rlc1-S36A* or the phosphomimetic versions *rlc1-S35D* and *rlc1-S35D S36D*, also grew very slowly in this medium (Figure 1D).”

“Conversely, expression of the phosphomimetic *rlc1-S35D* allele did not alter CAR dynamics and constriction rate when grown with glycerol (Figure 1F-G).”

(2) As far as I could tell, all data concerning the interplay between the two pathways (SAPK-For3 and PAK-Rlc1) are based on the rescue of the latter by expression of For3-DAD. In a simple scheme, this would put SAPK-For3 downstream of PAK-Rlc1; however, normal degradation of For3 in Rlc1-S35A cells (Figure 4A) rejects this model. I believe this point deserves to be emphasized in the manuscript. This leaves two other models: two independent pathways (Figure S6), or PAK1-Rlc1 being activated in response to reduced For3. These can be dissected by testing if For3-DAD expression suppresses Rlc1 phosphorylation.

Following the reviewer’s suggestion, we have used Phos-tag followed by Western blot analysis to explore whether Rlc1 in vivo phosphorylation at Ser35 is negatively affected by the *for3-DAD* allele. As shown in the new Figure 4—figure supplement 1E, Rlc1 Ser35 phosphorylation remained unchanged in cells expressing the *for3-DAD* allele compared to those expressing the wild-type allele, either during fermentation or respiration.

As suggested, we have included a new paragraph in the Results section of the revised manuscript describing the epistatic relationships between the SAPK-For3 and PAK-Rlc1 pathways during respiratory control of fission yeast cytokinesis:

“SAPK-For3 may act downstream of the PAK-Rlc1 cascade, as the cytokinesis and growth defects of *rlc1-S35A* and *pak1-M460G pak2∆* mutants are suppressed by *for3-DAD*. However, normal For3 degradation of For3 during respiration in *rlc1-S35A* cells (Figure 4A) excludes this possibility. Moreover, *for3-DAD* expression did not reduce Rlc1 phosphorylation at Ser35 in the presence of glucose or glycerol (Figure 4—figure supplement 1E), suggesting that Pak1/2-Rlc1 is not activated by changes in For3 levels. Thus, both the SAPK-For3 and PAK-Rlc1 pathways act independently and are critical for the successful completion of cytokinesis during respiration.”

(3) As stated in the above section, I strongly suggest that the effects of ROS and antioxidants in sty1∆ be examined.

We could not analyze CAR dynamics or the effect of antioxidants in a *sty1∆* background, because, unlike mutants lacking the upstream Wis1 (MAPK) or Wak1/Win1 (MAPKKs), *sty1∆* and *atf1∆* mutants (lacking the Sty1 downstream transcription factor Atf1) cannot grow on this carbon source (Reference 45: Zuin A et al. PLoS One. 2008;3(7):e2842). This phenotype is also shown in the new Figure 4—figure supplement 1C in the revised manuscript. The respiration-defective phenotype of *sty1∆* cells suggests that Sty1 may acts as a scaffold for recruiting the transcriptional machinery during respiration, independently of its activation status, similar to Hog1, the MAPK ortholog in *S. cerevisiae*.

In any case, *wis1∆* cells show identical phenotypes to those of the *sty1∆* mutant related to the negative control of For3 function, including the presence of thickened actin cables, increased actin cable-to-patch ratio during growth with glycerol (as shown in the new Figure 4—figure supplement 1A). Most importantly, Wis1 deletion increased the actin cable to patch ratio in *rlc1-S35A* cells (Figure 4—figure supplement 1A), suppressed their delayed cytokinesis and reduced constriction rate (Figure 4C-D), multiseptation (Figure 4—figure supplement 1B), and was able to restore cell growth in glycerol (Figure 4E). All this evidence confirms that the downregulation of For3 during respiration is strictly dependent on Sty1 activation.

We have included new explanatory text in the Results section of revised manuscript as follows:

“In agreement with previous observations [25], For3 levels increased in glucose-grown wild-type or *rlc1-S35A* cells lacking Sty1 activity by deletion of Wis1, the only MAPKK of the SAPK [44], and also in the presence of glycerol (Figure 4A-B). We could not analyze the cytokinetic and growth defects of *rlc1-S35A* cells during respiration in the absence of Sty1, because, unlike mutants lacking Wis1 or Wak1/Win1 (MAPKKs), *sty1∆* and *atf1∆* cells (lacking the Sty1 downstream transcription factor Atf1) are growth defective on this carbon source (Figure 4—figure supplement 1C) [45]. Nevertheless, similar to the *sty1∆* mutant [25], *wis1∆* cells exhibited thickened actin cables and an increased actin cable-to-patch ratio during growth on glycerol (Figure 4—figure supplement 1A). Furthermore, Wis1 deletion increased the actin cable- to-patch ratio in *rlc1-S35A* cells (Figure 4—figure supplement 1A), suppressed their delayed cytokinesis and reduced constriction rate (Figure 4C-D), multiseptation (Figure 4—figure supplement 1B), and restored cell growth in glycerol (Figure 4E).”

In addition, the revised Figure 4A shows a simplified scheme to assist readers unfamiliar with the details of For3 down-regulation by the SAPK pathway. Finally, and as suggested by the reviewer, the revised Figures 6B and 6C now include quantitative data showing that, as expected, CAR dynamics and constriction rates for *wis1∆* cells during respiration were not affected by the presence or absence of antioxidants (GSH). This is described in the Results section to read:

“Remarkably, 0.16 mM of the antioxidant tripeptide reduced glutathione (GSH) in the growth medium counteracted many effects of oxidative stress in *rlc1-S35A* and *myo2-E1* cells grown in glycerol. Changes included recovery of For3 levels (Figure 6A), the cable-to-patch ratio (Figure 6—figure supplement 1A), the timing of CAR assembly/contraction and constriction rate (Figure 6B-C), septation (Figure 6—figure supplement 1B) and growth (Figure 6D). Normal growth depended on For3 (Figure 6D). In contrast, in *wis1∆* cells GSH had no significant effect on the above-mentioned characteristics (Figure 6 B-D). Hence, respiration-induced oxidative stress reduces the nucleation of actin cables by formins and renders the execution of cytokinesis dependent on the phosphorylation of the Myo2 light chain Rlc1.”